

# An improved vertical correction method for the inter-comparison and inter-validation of Integrated Water Vapour measurements

Olivier Bock[1,2], Pierre Bosser[3], Carl Mears[4]

[1] Université de Paris, Institut de physique du globe de Paris, CNRS, IGN, F-75005 Paris, France
[2] ENSG-Géomatique, IGN, F-77455 Marne-la-Vallée, France
[3] Lab-STICC UMR 6285 CNRS / PRASYS, ENSTA Bretagne / HOP, F-29200 Brest, France
[4] Remote Sensing Systems, Santa Rosa, California, USA

*Correspondence to*: Olivier Bock (bock@ipgp.fr)

**Abstract.** Integrated Water Vapour (IWV) measurements from similar or different techniques are often inter-compared for calibration and validation purposes. Results are usually assessed in terms of bias (difference of the means), standard deviation of the differences, and linear fit slope and offset (intercept) estimates. When the instruments are located at different elevations, a correction must be applied to account for the vertical displacement between the sites. Empirical formulations are traditionally used for this correction. In this paper we show that the widely-used correction model based on a standard, exponential, profile

for water vapour cannot properly correct the bias, slope, and offset parameters simultaneously. Correcting the bias with this model degrades the slope and offset estimates, and vice-versa. This paper proposes an improved correction model which overcomes these limitations. The model uses a multi-linear regression of slope and offset parameters from a radiosonde climatology. It is able to predict monthly parameters with a root-mean-square error smaller than 0.5 kg m$^{-2}$ for height differences up to 500 m. The method is applied to the inter-comparison of GPS IWV data in a tropical mountainous area and

to the inter-validation of GPS and satellite microwave radiometer data. This paper also emphasizes the need for using a slope and offset regression method that accounts for errors in both variables and for correctly specifying these errors.

## 1 Introduction

Water vapour plays a key role in many meteorological processes and in the hydrological cycle of the Earth's atmosphere. Because it is extremely heterogeneous and variable, many operational and research observing techniques have been developed

over the years to sense its horizontal, vertical and temporal variability. Among the various high-performing techniques, one may cite in-situ measurements with radiosonde balloons and remote sensing techniques exploiting different domains of the electromagnetic spectrum, namely Fourier Transform Infrared Radiometers (FTIR), near-infrared, visible, and ultraviolet radiometers and spectrometers, as well as microwave radiometers (MWRs) and microwave measurements from Global Navigation Satellite Systems (GNSS). Integrated Water Vapour (IWV) measurements from ground-based and space-based

platforms are often compared to assess each other's accuracy, e.g. detect biases and/or long term drifts (Bokoye et al., 2003;





Morland et al., 2006a,b; Bock et al., 2007; Bokoye et al., 2007; Morland et al., 2009; Sussmann et al., 2009; Bedka et al., 2010; Palm et al., 2010; Schneider et al., 2010; Vogelmann et al., 2011; Buehler et al., 2012; Cimini et al., 2012; Bock et al., 2014; Van Malderen et al., 2014; Courcoux and Schröder, 2015; Schröder et al., 2016; Bock et al., 2021), as well as for inter-calibration purposes, e.g. adjusting biases from instruments on successive space-based platforms (Du et al., 2014; Mears et al.,

2015; Wentz, 2015; Bennartz, 2017; Mears et al., 2018; Ho et al., 2018; Schröder et al., 2019). In this context, it is frequent that IWV measurements from sites at different elevations need to be compared (e.g., Bock et al., 2005; Morland et al., 2006b; Buehler, 2012; Van Malderen et al., 2014). Because, the water vapour concentration in the atmosphere is decreasing by several orders of magnitude between the surface and the upper troposphere, a vertical correction is required to conform the measurements from sites at different altitudes, or between point observations and aerial averages such as derived from

atmospheric models (Bock et al., 2005, 2007; Morland et al., 2006a; Buehler et al., 2012; Bock et al., 2014). While many studies have recognized that a height difference results in a systematic difference (bias) in the IWV measurements, few have applied a correction, and even fewer have recognized that the height difference also impacts the linear fit slope and offset estimates. To our knowledge, only Bock et al., 2005, Morland et al., 2006a, b, Buehler et al., 2012, and Van Malderen et al., 2014, addressed these points. Van Malderen et al., 2014, experienced that applying a scaling factor for correcting the bias is

degrading the slope estimate. Buehler et al., 2012, analysed the impact of height difference on the slope estimate using radiosonde data and proposed to use this estimate to correct the IWV data. We will follow a similar methodology in this paper with an improved model. Among the correction models that have been proposed by various authors, two approaches have been traditionally used. The first, and most widely used one, is based on a proportional correction term, with takes its roots in the assumed exponential decrease of water vapour concentration with height. This model is described in Appendix A. It makes

use of the assumption of a constant vertical decay rate of water vapour, $\gamma$. At least three studies have applied this correction model with nearly similar experimental values for $\gamma$; namely, Bock et al., 2005, proposed $\gamma = 4 \cdot 10^{-4}$ m$^{-1}$ for the Alps, Morland et al., 2006b, proposed $\gamma = 4.7 \cdot 10^{-4}$ m$^{-1}$ also for the Alps, and Buehler et al., 2012, proposed $\gamma = 3.5 \cdot 10^{-4}$ m$^{-1}$ for Antarctica. These models have been claimed by their authors to be valid for height differences up to 500 m. The second approach, proposed by Mears et al., 2015, is based on the observed sea surface temperature and a constant relative humidity

of 80%. They applied this model for the inter-comparison of satellite-based MWR measurements with ground-based Global Positioning System (GPS) stations with elevations usually less than 100 m, and one exceptional case above 500 m for which it still worked well. Both approaches have been shown to provide acceptable reductions of the differential IWV biases.

In this paper, we show that the exponential correction cannot simultaneously achieve a proper correction for the bias and for the slope and offset parameters. To overcome this limitation, we propose an improved vertical correction method based

on multi-linear regression from a radiosonde climatology. Another aspect discussed in this paper is the impact of errors in both variables on the slope and offset estimates. Contrary to trend estimation, where a physical variable is regressed on time (a quantity known with negligible error), the linear regression between two measurements which are both subject to errors requires a more subtle estimation method. Indeed, it has been shown that in this situation, the ordinary least-squares (OLS) regression leads to biased estimates of the slope and offset (Draper and Smith, 1998). This problem has clearly been overlooked





in the aforementioned IWV intercomparison literature. This may be at least one reason for the variety of slope results found in these studies (with slope being either bigger or smaller than one). Comparing slope and offset results from different studies, such as done, e.g., in Buehler et al., 2012, may therefore be hazardous as different and sometimes wrong regression methods have been used. Only few of these studies stated explicitly that they used a regression method accounting for errors in both variables (e.g., Buehler et al., 2012; Bock et al., 2014, 2021). Using such a method also poses the problem of correctly

specifying the uncertainties in both variables. Lack of such information for some of their data sets led Buehler et al., 2012, to apply OLS regression and to state that constant error estimates do not affect the regression results, which is wrong (see Appendix C). Instead, Bock et al., 2014 and 2021, used approximate error estimates, e.g. 5% or 10% for radiosonde or satellite data, rather than assuming no errors in the $x$ variable. In this paper, we use the three-way error analysis of O'Carroll, et al., 2008, to specify the uncertainties of our data sets, and the regression method of York et al., 2004, which accounts for errors in

both variables.

        Section 2 of this paper reviews the impact of a height difference on the bias, slope, and offset estimates in the case of an idealized exponentially decaying water vapour density profile and in the case of a real atmosphere observed by radiosondes. The similarities and differences implied by two types of profiles are highlighted. Section 3 proposes an improved vertical correction method based on a multiple linear regression approach, instead of using one single $\gamma$ parameter as done in past

studies. The method builds on a climatology derived from radiosonde data. In Section 4 we discuss two application examples where IWV measurements from a network of GPS stations in a tropical mountainous area are to be inter-compared and used for the inter-validation with collocated satellite MWR measurements. Both applications make use of the new method and the derived radiosonde climatology. Section 5 discusses further applicability of the method and concludes.

**2 Variation of bias, slope, and offset parameters as a function of height difference**

**2.1 Idealized case of an exponentially decaying water vapour density profile**

Before analysing the results from real data, it is instructive to consider the idealized case of a water vapour density profile decaying exponentially with height (Eq. (A1)). This model has often been used to describe the state of the mean atmosphere (e.g., ITU, 2017) and is related to the notion of water vapour scale height (see Appendix A).

        Let us consider the situation of two instruments, A and B, located at sites with different heights, $h_A$ and $h_B$, which are

observing IWV in an idealized atmosphere described by Eq. (A1). In the absence of any instrumental bias and noise, the IWV observations are related by:

$$IWV_B = IWV_A \times \exp(-\gamma(h_B - h_A)) , \qquad (1)$$

where $\gamma > 0$ is the vertical decay rate of water vapour, which is related to the water vapour scale height by the relation $H_v = 1/\gamma$. If $h_B > h_A$ we have $IWV_B < IWV_A$, i.e. the IWV content at higher altitude is lower than the IWV content at lower altitude.

If the observations from station A and B are directly compared without any correction, we will observe a negative bias, $\Delta=$





$IWV_B - IWV_A < 0$, a slope smaller than one, $\alpha < 1$, and a null offset, $\beta = 0$, where the slope and offset are estimated by a linear regression using the model $y = \alpha x + \beta$, with $y = IWV_B$ and $x = IWV_A$ (see Appendix B).

To be a bit more general, we can assume that the observations contain some amount of random noise and consider that we have $n$ pairs of observations, $(x_i, y_i)$, $i = 1..n$, from which the bias, slope and offset parameters are estimated. The bias writes:

$$\Delta = \frac{1}{n}\sum_{i=1}^{n}(y_i - x_i) = \mu_y - \mu_x , \tag{2}$$

where $\mu_x$ and $\mu_y$ denote the sample means of the two data series. From Eq. (1), and introducing $f(\Delta h) = \exp(-\gamma \Delta h)$, with $\Delta h = h_B - h_A$, the bias can be expressed as:

$$\Delta = -\mu_x \times [1 - f(\Delta h)] = -\mu_x \times [1 - \exp(-\gamma \Delta h)] \approx -\mu_x \gamma \Delta h, \tag{3}$$

The first right-hand side (rhs) will be used later to describe the more general atmospheres. The second rhs is valid only in the case of the exponentially decaying water vapour profile. It expresses that the bias is proportional to the mean IWV content at the reference site, $\mu_x$, and that it is negative given that $\gamma > 0$ and $\Delta h > 0$. The last rhs, is the approximate relation valid for a thin layer (typically, $|\Delta h| < 200$ m, see Appendix A), and expresses that, to the first order, the bias is proportional to $\mu_x$, $\gamma$, and $\Delta h$. This last expression has been used in past studies (e.g., Bock et al., 2005; Buehler et al., 2012) to estimate the vertical moisture decay rate, $\hat{\gamma} = -\Delta \times (\mu_x \Delta h)^{-1}$, and to correct IWV observations for the height difference between sites. The slope and offset parameters estimated from the linear regression establish a second relation between $\mu_x$ and $\mu_y$ given by Eq. (B3), which can be rewritten in the case of the exponential water vapour profile as: $\mu_y = \alpha \mu_x + \beta = \mu_x \times f(\Delta h)$. Since this relation must be valid for any $\mu_x$, it comes out that:

$$\alpha = f(\Delta h) = \exp(-\gamma \Delta h) \approx 1 - \gamma \Delta h, \tag{4a}$$

$$\beta = 0. \tag{4b}$$

The second rhs of Eq. (4a) expresses that $\alpha < 1$, given that $\Delta h > 0$, and the third rhs that, to the first order, $\alpha$ decays linearly with the height difference, with a rate equal to $\gamma$.

Figure 1 illustrates the main characteristics of the bias and slope variations with height in the case of the idealized exponential water vapour profile, along with the asymptotic limits and the thin layer linear approximations. It is important to note that not only the bias is changing when the depth of the atmospheric layer $\Delta h$ is changing ($|\Delta|$ is increasing when $|\Delta h|$ is increasing), but also the slope ($|\alpha - 1|$ is increasing when $|\Delta h|$ is increasing).

Equations (3) and (4a) recall also that both the bias and slope parameters depend on the atmospheric profile through the $\gamma$ parameter which may be of relatively local nature and may thus be changing from one region to another and varying with time, e.g. seasonally. Moreover, in real atmospheres, the vertical distribution of water vapour is expected to be more complex than represented by an exponential model with a constant vertical decay rate.



## 2.2 Real case from radiosonde observations

Figure 2 illustrates the monthly mean water vapour profiles observed by a tropical radiosonde station (Le Raizet, Guadeloupe, France, WMO code 78897), over the year 2020. It can be seen that the water vapour is decaying approximately exponentially,

although the vertical decay rate is not strictly constant as a function of height and time. This model is nevertheless reasonable in the lower troposphere, e.g., from the surface up to a height of 2 km (Fig. 2b). In this altitude range, we expect Eq. (1) to be a good approximation of the vertical variation of IWV. Figure 2c illustrates the link between $x = IWV(h_s)$ and $y = IWV(h_s + \Delta h)$, where $IWV(h_s) = \int_{h_s}^{\infty} \rho_v(h) dh$ and $IWV(h_s + \Delta h) = \int_{h_s + \Delta h}^{\infty} \rho_v(h) dh$, and where $\rho_v(h)$ is the observed radiosonde water vapour profile, $h_s$ is the station height, and $\Delta h$ is varied between 200 m and 1000 m by step of 200 m. For each layer, $\Delta h$,

the points $(x, y)$ align roughly on a straight line. For $\Delta h = 200$ m, the line is closest to the 1:1 line (shown in grey) and the scatter around the best fit line is the smallest (RMSE = 0.224 kg m$^{-2}$), while for $\Delta h = 1000$ m, the line is farthest from the 1:1 line and the scatter around the best fit line is the largest (RMSE = 0.976 kg m$^{-2}$). It is interesting to note that for a given layer, $\Delta h$, the points remain close to the straight line throughout the year, despite the quite large seasonal excursion in IWV shown by the different colours in the figure. The data points for March are shown as light blue dots and the data points for

September as orange dots. These two months show the smallest and largest mean IWV values, $\mu_x$, of 33 kg m$^{-2}$ and 51 kg m$^{-2}$, respectively.

Figure 3 shows the variations of the bias, offset, and slope parameters fitted from these data, as a function of $\Delta h$. The bias (Fig. 3a) and the fractional bias $(\Delta/\mu_x)$ (Fig. 3d) follow reasonably well the exponential decay predicted by the second rhs of Eq. (3), but the lines do not actually align perfectly from one month to another, because of the small seasonal variations

in the humidity profile. The monthly variation is even more visible in the slope and offset plots (Figs. 3b and c). However, each monthly curve for the slope may be reasonably well modelled by an exponentially decaying function described by the second rhs of Eq. (4a). Regarding the offset, the purely exponentially decaying water vapour profile predicts $\beta = 0$, which is clearly not verified in the real atmosphere. However, all three parameters together follow the relationship described in Appendix B, i.e. $\alpha < 1$ and $\Delta < 0$ implies that $\beta > \Delta$. Figure 3c, shows that $\beta$ actually follows closely the variation of $\Delta$ as a

function of $\Delta h$, while verifying $\beta > \Delta$. Figures 3b and 3c, also recall that the slope and offset estimates are correlated to each other, i.e. higher slopes are associated with smaller offsets (Walpole et al., 2012). Figures 3e and f show the standard errors of the $\alpha$ and $\beta$ parameters estimated by OLS, given by Eqs. (C4) and (C5a, b). They are increasing with $\Delta h$ as expected from the increased scatter of the post-fit residuals (Fig. 2c). In the next Section we will establish a model describing the behaviour of $\alpha$ and $\beta$ as a function of $\Delta h$ that will be used to correct the observations $x$ made at a height $h_A$ to conform to the observations $y$

made at height $h_B = h_A + \Delta h$.





## 3. Derivation of an empirical correction model from radiosonde observations

In the previous Section we have seen that the bias, $\Delta$, and the slope, $\alpha$, are dependent on the depth, $\Delta h$, of the layer between the two considered IWV observations. Especially, $|\Delta|$ and $|\alpha - 1|$ are both increasing when $|\Delta h|$ is increasing, both in the real and in the idealized, exponentially decaying, atmosphere. Whereas the offset $\beta = 0$ in the idealized atmosphere, it is generally
$\beta \neq 0$ in the real atmosphere. The main difference between the idealized and real atmospheres is that the vertical moisture decay rate $\gamma$ is dependent on the height (and time) in the latter, whereas it is by definition constant in the former (although a time variation could also be modelled in the idealized atmosphere). We must thus derive a correction formula based on a more complex model than just a constant $\gamma$. Moreover, a pure rescaling correction, $x_c = f_c(\Delta h) \times x$, as discussed in Appendix B, does not allow to correct simultaneously the bias and slope, and does not change the offset. Instead, we propose to use a linear
correction model such as expressed by Eqs. (B8) and (B9). Therefore, we need good estimates for both $\alpha$ and $\beta$, which are generally not known at the location and time of interest but may be derived from a climatology. Hereafter, we propose to use high-resolution radiosonde observations to derive such a climatology.

The proposed approach is to model the slope and offset with two independent functions of $\Delta h$:

$$-\log(\alpha) = A(\Delta h), \tag{5a}$$

$$\beta = B(\Delta h), \tag{5b}$$

which are represented by polynomials:

$$A(\Delta h) = \sum_{i=1}^{p} a_i \Delta h^i \tag{6a}$$

$$B(\Delta h) = \sum_{i=1}^{q} b_i \Delta h^i \tag{6b}$$

Note that the polynomials have no intercepts in order to satisfy the constraints $A(0) = 0$ and $B(0) = 0$. Figures 3b and 3c
suggest that the order of the polynomials does not need to be very high. For example, coefficient $a_1$ can be identified with the vertical moisture decay rate, $\gamma$, in analogy with Eq. (4a). The higher order terms help to model the deviations from linearity observed in Fig. 3b and 3c.

The estimates of the polynomial coefficients, for each of the two models, are derived by a linear regression method, according to the generic linear model equation: $\mathbf{z} = \mathbf{X}\boldsymbol{\theta}$, where $\mathbf{z}$ is the vector of dependent variables, $\mathbf{X}$ the design matrix, and
$\boldsymbol{\theta}$ the vector of parameters (Walpole et al., 2012). The elements, $z_k$, of vector $\mathbf{z}$ correspond either to the observed slope values, $-\log(\alpha_k)$, or to the offset values, $\beta_k$, for the different layers, $\Delta h_k$, $k = 1..m$. The elements of the design matrix are $X_{ik} = (\Delta h_k)^i$, and the parameters $\theta_i = a_i$, with $i = 1..p$, in the case of the slope model, and $\theta_i = b_i$, $i = 1..q$, in the case of the offset model. Note that here we estimate the slope and offset coefficients independently of each other. Another approach might be to consider both variables simultaneously in a multivariate linear regression (Christensen, 2001) which is possible here
since both variables are described by similar functional models, (6a) and (6b). A few tests of this approach revealed that both the estimates and their standard errors were identical to the monovariate solutions. So we decided to stay with the monovariate linear regression approach which is simpler to implement and faster to run.





The quality of the fitted models depends on the number of observations, $m$, and the choice of the polynomial orders, $p$ and $q$. Indeed, larger layers would require to include higher order terms to adequately fit the deviations from linearity. The

number of observations depends on the vertical sampling of the radiosonde profiles. Since we are using high-resolution radiosonde data, we can set a regular vertical sampling of $\Delta h$=25 m, i.e. $\Delta h_k = k \times \Delta h$. Considering two different maximal thicknesses of $\Delta h_m = 500$ m and $\Delta h_m = 1000$ m, this leads to $m$=20 and $m$=40, respectively.

The order of the polynomials can be either fixed to predetermined values or determined automatically by a stepwise linear regression method (Hocking, 1976). The stepwise regression selects the best model by adding/removing terms to/from

the model. The selection can be based on the $p$-value of the $F$-statistic associated to the change in the sum of squared errors (SSE) that results from adding or removing a term. Other types of criteria such as the Akaike Information Criterion (AIC), the Bayesian Information Criterion (BIC), or the adjusted coefficient of determination $\bar{R}^2$, can be used as well (Draper and Smith, 1998). A few trials with different values for $p$ (resp. $q$), revealed that all the aforementioned criteria (SSE, AIC, BIC, and $\bar{R}^2$) lead to very consistent results, and that the quality of the model is generally improved when $p$ (resp. $q$) is increased. However,

we also noticed that when $p > 5$ (resp. $q > 5$), the regression failed due to poor conditioning of the normal matrix, $X^T X$. We consequently limited the regression to maximum orders $p = 5$ (resp. $q = 5$). The SSE criterion was used with the following limits for the $p$-values: when $pval < 0.05$, the term is added during the forward step, while when $pval > 0.10$ the term is removed during the backward step. This method is, e.g., implemented in the *stepwiselm* function available in Matlab (2017).

Another aspect of the implementation of the linear regression method is whether we consider the data as

homoscedastic (the observations have constant variance) or heteroscedastic (the observations have different variance). Figures 3e and 3f suggest that a heteroscedastic model is plausible: the standard errors, $\sigma_{k,\alpha}$ and $\sigma_{k,\beta}$, of the "observations", $\alpha_k$ and $\beta_k$, are generally increasing with $k$. Heteroscedasticity can be simply accounted for by specifying a diagonal weight matrix, $W$, where the diagonal elements are $W_{kk} = w_{k,\alpha}$ for the slope and $W_{kk} = w_{k,\beta}$ for the offset, which are computed here from the standard errors, i.e., $w_{k,\alpha} = (\sigma_{k,\alpha}/\alpha_k)^{-2}$ and $w_{k,\beta} = (\sigma_{k,\beta})^{-2}$. Note that the relative standard error is used in the case of

the slope, because we use $\log(\alpha)$ and not $\alpha$ in the regression.

We conducted a large number of trials for different values of the model parameters: $p = 1..5$, $q = 1..5$, $m$=20 and $m$=40 (maximum layer depths of 500 and 1000 m), weighted or un-weighted regression, and different data sets: monthly or yearly input data (i.e. $\alpha$ and $\beta$ fitted month by month or from a full year of radiosonde profiles). We also compared the regression results from different radiosonde stations to assess the robustness of the method as well as the spatial variability of

the fitted parameters. The results from the different trials were inter-compared on the basis of two quality criteria: the standard error of the regression, also called the root mean square error (RMSE), and the standard error (SE) of the estimates (Draper and Smith, 1998). The RMSE quantifies the dispersion of the observed values, $z_k$, around the predicted values, $\hat{z}_k$, adjusted for the degrees of freedom:

$$s_e = \left[\frac{1}{m-p}\sum_{k=1}^{m} \hat{e}_k\right]^{1/2} \tag{7}$$



where $\hat{e}_k = z_k - \hat{z}_k$ is the prediction error and $m - p$ is the degrees of freedom in the case of the slope ($m - q$ in the case of the offset). The SE of the estimates is obtained from the variance-covariance matrix $\boldsymbol{Q}$:

$$SE(\theta_i) = \sigma_e \times (Q_{ii})^{1/2} \tag{8}$$

where $\sigma_e$ is the standard deviation of the errors in the "observations", an estimate of which is given by $\hat{\sigma}_e = s_e$, and $\boldsymbol{Q}$ depends only on the regressors, $\boldsymbol{X_i}$, and the weights, $W_{kk}$. In the case of the OLS, $\boldsymbol{Q} = (\boldsymbol{X^T X})^{-1}$, while in the case of the weighted

least-squares (WLS), $\boldsymbol{Q} = (\boldsymbol{X^T W X})^{-1}$. It is straightforward to show that in the case of OLS, a simple polynomial model such as expressed by Eq. (6a) and limited to the order $p =1$, leads to $(X_{11})^2 \approx \frac{m^3}{3} \Delta h^2$ when the terms in $m^2$ and smaller order are neglected. This result indicates that the SE of the parameters varies as $m^{-3/2}$. We may thus expect some benefit from performing the regression over more elevated layers, e.g. with $m$=40 compared to $m$=20, although the final SE also depends on the standard error of the regression, $s_e$, which may be increasing when more elevated layers are included.

230         The results obtained from the trials are summarized below:

- The RMSE is decreased when the order of the model ($p$ or $q$) is increased. This result is expected as a higher order model better fits the real data.
- The RMSE is increased with WLS compared to OLS. This is a statistical property of OLS compared to WLS, i.e. OLS generally better fits the original data than WLS (Draper et al., 1998).

- The SE of the estimates is increased when the order of the model is increased. This result is expected from the fact that more parameters are estimated with the same number of observations.
- The SE of the estimates is decreased with WLS compared to OLS. This result is expected because OLS is no longer the best linear unbiased estimator when the errors in the data are not equal (Draper et al., 1998).
- Both the RMSE and the SE are increased when more elevated layers (up to 1000 m compared to 500 m) are considered,

despite the increase in the number of observations ($m$=40 compared to $m$=20).

The above results were found valid for both variables ($z_k = -\log(\alpha_k)$ and $z_k = \beta_k$), both time samplings (monthly and yearly), and also the different stations considered. They suggest to use preferably high order polynomials (e.g., $p = q = 5$), WLS estimation, and a limited vertical extent of the regression (e.g. 500 m).

Figure 4 shows the estimates for parameters $a_1$ and $b_1$, to illustrate the variability in time and space, at three stations

located in the Caribbean region (78526 is located 531 km to the north-west from 78897, on Puerto-Rico island, and 78954 is located 417 km to the south from 78897, on Barbados island). The temporal variations at each of the three sites are significant (compared to the error bars) but correlated between the sites. These results indicate that: (i) it may be preferable to use monthly regression coefficients rather than yearly, and (ii) the radiosonde climatology derived from one site may be applied to distant sites to some extent (e.g. a few hundreds of kilometres apart). We further checked the first point by analysing the bias after

correction, $\Delta_c = \mu_y - \mu_{x,c}$, as a function of the height difference, $\Delta h$, for monthly and yearly coefficients. The correction model





is expressed by Eqs. (B8) and (B9), where $f_c$ and $g_c$ are derived from the predicted values for $\hat{\alpha}$ and $\hat{\beta}$ according to Eqs. (5) and (6):

$$\hat{f}_c = \exp(-\sum_{i=1}^{p} \hat{a}_i \Delta h^i) \tag{9a}$$

$$\hat{g}_c = \sum_{i=1}^{q} \hat{b}_i \Delta h^i \tag{9b}$$

Figure 5 compares the IWV correction errors for monthly and yearly coefficients, $\hat{a}_i$ and $\hat{b}_i$, fitted by WLS, with $p = q = 5$. Larger dispersion is clearly observed with the yearly model, with significant seasonal variation with an amplitude increasing with $\Delta h$.

Figure 6 gives more information for the model with monthly coefficients. The monthly mean bias remains negligible, $|\Delta_c| < 0.02$ kg m$^{-2}$, and independent of the height difference, which demonstrates that the correction model is well fitted. The
standard deviation is increasing with the height difference up to 0.5 kg m$^{-2}$ when $\Delta h = 500$ m (this quantifies the dispersion observed in Fig. 5a). The slope and offset after correction are significantly improved and get very close to the ultimate objective ($\alpha_c = 1$ and $\beta_c = 0$). In conclusion, the proposed correction method is able to reduce almost perfectly the impact of the height difference on the IWV observations and achieve $\Delta_c \approx 0$, $\alpha_c \approx 1$, and $\beta_c \approx 0$, when monthly coefficients are used.

## 4. Applications

### 4.1 GPS vs. GPS inter-comparison

We will consider here the case of the permanent GPS network in Guadeloupe analysed by Bock et al., 2021. It is composed of 15 stations located on 4 islands, in a region bounded by 62°W - 61°W and 15.75°N - 16.75°N. The station elevations range from 1 m to 418 m (see Table 1). The ultimate goal of this inter-comparison is to determine the consistency between the IWV measurements from these GPS stations and to check for biases and non-linearities. Therefore, the IWV data need to be
corrected for the height differences. We will here consider both the simple scaling factor model, based on (A5), and the new model based on (9a) and (9b). These correction models will be referred to as v1 and v2 in the following, and the uncorrected data will be denoted v0. The model coefficient in v1 is taken to $\gamma = 4 \cdot 10^{-4}$ m$^{-1}$ consistent with Bock et al., 2007, for the tropics. In v2, we will use the coefficients determined from the radiosonde climatology derived in the previous section from the radiosonde station 78897 which is located on the Guadeloupe island, close to the GPS station ABMF. The bias, slope, and
offset parameters derived from the inter-comparisons for the different data versions will be denoted $\Delta_v$, $\alpha_v$, and $\beta_v$, with $v = 0, 1, 2$, respectively.

Figure 7 shows the results of the inter-comparison of two stations at very different elevations: ABMF, $h_A = 15$ m, and HOUE, $h_B = 418$ m. It is seen that the initial bias of $\Delta_0 = -7.29$ kg m$^{-2}$, is quite large but consistent with the values predicted from the radiosonde data (Fig. 3) for such a large height difference. Both correction models reduce significantly the bias,
although v1 has some residual bias, $\Delta_1 = -2.35$ kg m$^{-2}$, whereas v2 achieves $\Delta_2 = -0.50$ kg m$^{-2}$, i.e. almost perfect



correction. Figure 7 also compares the slope and offset results estimated by two different regression methods: Figs. 7*a*, *b*, and *c* used the OLS method, i.e. assuming no errors in the *x* variable, and Figs. 7*d*, *e*, and *f* used the York et al., 2004, method. With the latter method, the formal errors provided by the GPS data processing software were used as "obsevation errors", after a rescaling by factor of 5 to be consistent with the traditionnally assumed uncertainty of 1.5 kg m$^{-2}$ for GPS IWV data (Bock

et al., 2021). The initial slope and offset amount to $\alpha_0 = 0.92$ and $\beta_0 = -4.63$ kg m$^{-2}$ with the OLS estimator, and $\alpha_0 = 0.97$ and $\beta_0 = -6.34$ kg m$^{-2}$ with the York estimator. The latter values are more in line with the values found from the radiosonde data (Fig. 3) and predict a higher slope. It is well known that the OLS slope estimator is biased low (towards zero) when the *x*-variable contains random errors (Edland, 1996). This feature is clearly observed with all three data versions shown in Fig. 7. The results also verify the relationship between bias, slope, and offset sketched in Fig. B1, whatever the estimator. After

correction with model v1, the slope becomes $\alpha_1 = 1.08$ with the OLS estimator and 1.15 with the York estimator, while correction with model v2 achieves $\alpha_2 = 0.95$ with the OLS estimator and 1.01 with the York estimator. Both estimators find that model v1 over-corrects slightly the data ($\alpha_1 > 1$). On the other hand, v2 performs much better, and achieves almost a perfect slope ($\alpha_1 \approx 1$) with the York estimator. Regarding the offset, we see that the value is unchanged with model v1, as predicted from (B7), whereas model v2 achieves nearly perfect correction ($\beta_2 \approx 0$). These results are highly consistent with

those found in Section 3 from the radiosonde data. Regarding the initial question, we can state the IWV measurements from stations HOUE and ABMF are fairly consistent after the vertical correction. The residual bias and offset after correction are fairly within the error bars of the technique (Bock et al., 2013; Ning et al. 2016).

Figure 8 presents the results for 105 inter-comparisons made of pairs of stations from the set of 15 stations of this network ordered by positive height differences, $\Delta h > 0$. The plots compare the bias, offset and slope for the uncorrected (v0)

and the corrected (v1 or v2) data. Only the results from the York estimator are shown here. As expected, the uncorrected results show a general tendency towards larger negative biases, decreasing slope and larger negative offset when $\Delta h$ is increased. There are, however, some exceptions, namely the comparisons involving station CBE0 (altitude of 374 m), for which the biases and offsets are slightly less negative, and the slopes are slightly farther from one, than observed with the other stations, especially compared to station HOUE which is located higher (418 m) and should thus have more pronounced effects. After

correction with model v1, the biases and slopes are globally improved for all comparisons, while the offsets are unchanged, as expected with this model. The mean bias is reduced from $-1.67$ kg m$^{-2}$ to $-0.24$ kg m$^{-2}$ but some bias remains in the higher altitude inter-comparisons involving HOUE (Fig. 8a). In contrast, model v2 achieves a better bias correction for HOUE (Fig. 8d). The results with model v2 also confirm the bias in the CBE0, which was already suspected from the uncorrected data. The problem with station CBE0 is further confirmed by the slope analysis, with model v2 indicating $\alpha_2 < 1$ for these inter-

comparisons (Fig. 8e), and large positive offsets (Fig. 8f). The correction with model v1 is not able to lead to these conclusions because the slopes are globally over-corrected for many stations (Fig. 8b) and the offsets are unchanged (Fig. 8c). Figures 8e and f also detect scale errors and anomalous offsets for a number of other inter-comparisons, namely when $\Delta h$ is close to zero.

Figure 9 provides further insight into the consistency between stations, with significance tests computed according to the *t*-statistics given in Appendix C. It is evident that CBE0 has an anomalous positive IWV bias about 2 kg m$^{-2}$ compared to



all other stations (Fig. 9a, red curve, well above the other curves), a slope too low (Fig. 9b, red curve below the other curves), and a too large offset (Fig. 9c). Figures 9b and 9c reveal a second outlying station, BOUL, with a too high slope (Fig. 9b, light blue curve, about 0.12 above the other stations) and a too low offset (Fig. 9c). These anomalies could neither be detected from the uncorrected data, nor from the data corrected with model v1. Further investigation is needed to understand the issues in the IWV estimates for these two stations. Table 2 reports the median and the smallest absolute values for each station. Apart from

stations CBE0 and BOUL, all other stations have median biases smaller than $\pm 0.53$ kg m$^{-2}$, median slopes in the range 0.97-1.02, and median offsets smaller than $\pm 0.77$ kg m$^{-2}$. These numbers demonstrate a very good consistency between IWV measurements retrieved from the different GPS stations of this network. The dispersion of results is believed to be due to station-dependent errors. The smallest absolute values quantify the best agreement between nearby GPS stations, which is $<$ 0.1 kg m$^{-2}$ between all stations, except CBE0 which has a large bias, and BOUL which has a slope significantly different from

325     1.0.

### 4.2 GPS vs. MWR satellite inter-validation

GPS and MWR measurements of IWV are often used together for the inter-validation of the two techniques (Bock et al., 2007; Mears et al., 2015; Wentz, 2015; Ho et al., 2018). Microwave radiometer measurements are adversely affected by rain whereas GPS measurements are not. On the other hand, the GPS IWV estimates have uncertainties linked with data processing models

and conversion from propagation delay to IWV (Bock et al., 2013; Ning et al., 2016; Bock et al., 2021). The intercomparison of both types of measurements is thus instructive for detecting and quantifying their mutual uncertainties.

            Microwave radiometer measurements are traditionally made over the world's oceans where they achieve their highest accuracy. The intercomparison with GPS measurements is thus possible only for coastal stations and stations located on small islands. Although the MWR data are missing over land and over island's footprint, due to "land contamination", the high

resolution (0.25°x0.25°) of the RSS v7.0 data set used here (Mears et al., 2015) allows to get enough valid measurements for comparison with the GPS stations on the Guadeloupe islands discussed in Sect. 4.1. Table 3 shows the mean distance between the GPS stations and the nearest MWR satellite grid-points within the 7 x 7 pixels surrounding each station. On average over all stations, the mean distance is 33.6 km for AMSR2, 82.3 km for F18, 26.9 km for GMI, and 37.6 km for Windsat. The difference in distance is due to the difference of footprints of the satellite instruments, F18 having the largest footprint (69 km

x 43 km), and GMI the smallest (18 km x 11 km). For the intercomparison, MWR IWV data from the 7 x 7 pixels are interpolated to the location of the GPS sites by a Delaunay triangulation method (Press et al., 2007) and corrected vertically using the same method as for the GPS – GPS comparison discussed in Section 4.1. The height difference is here equal to the height of the GPS station, since the MWR data are valid on the mean sea level. The bias, slope, and offset parameters are derived as in the GPS – GPS comparison. The regression with the York et al., 2004, method, needs to specify correctly the

uncertainties of the measurements from the two data sets, or at least to represent correctly the ratio of their mean uncertainties (see Appendix C). As mentioned above, the formal error rescaled by a factor of 5 is used for GPS. For MWR, we surmise that the horizontal interpolation from the gridded data will introduce some representativeness difference with the GPS point





measurements that we should take into account. We first computed the standard deviation of all valid IWV values from the 7 x 7 pixels. It amounted to ~ 2.2 kg m$^{-2}$ on average over all sites and satellites. This values seemed too high to be used directly

as a measure of uncertainty of the MWR data. However, the variations over time of the standard deviation are thought to correctly reflect the changes in the local atmospheric state, weather conditions, and measurement noise. In a second step, we made a three-way error analysis between GPS, MWR, and ERA5 IWV data, following O'Carroll et al., 2008. This was done with the GPS station BCON, on Barbados island (Bock et al., 2021). For this station, the number of valid MWR pixels was higher than at all other sites of Caribbean GPS network, with an average of 46 valid pixels out of 49. This comparison is thus

believed to provide a good estimate of the precision of the MWR data with negligible representativeness errors. We found the following standard error estimates for the three data sets: $\sigma_{GPS}$ = 1.06 kg m$^{-2}$; $\sigma_{MWR}$ = 0.67 kg m$^{-2}$; $\sigma_{ERA5}$ = 1.82 kg m$^{-2}$, for AMSR2. Nearly similar values were found for the other satellites. According to these numbers, the GPS IWV data are slightly noisier than the MWR data, which seems plausible, although the MWR and ERA5 standard errors might be slightly underestimated given that MWR radiances are assimilated into ERA5, i.e. errors are correlated. Finally, we rescaled the GPS

formal errors and the MWR standard deviations to match these three-way error values, on average, for each satellite. The resulting "measurements errors" were then used in the York fit.

Figure 10 shows the results of the GPS – MWR comparisons, where the bias, slope, and offset parameters were retrieved for the whole year 2020. The number of collocations here is much smaller than for the GPS – GPS comparisons (between 200 and 400 for the GPS – MWR comparisons compared to more than 4000 for the GPS – MWR comparisons over

the full year). The median GPS – GPS results, determined as in Sect. 4.1, are superposed to emphasize the high correlation with the GPS – MWR intercomparisons (the Pearson correlation coefficients reported in each plot). Regarding, the biases, especially, the variations from station to station are about ± 0.5 kg m$^{-2}$ (if we except CBE0) from both intercomparisons. They are thought to be GPS station-specific errors (due to, e.g., multipath and/or field of view limitations). The large bias in CBE0 is confirmed with the MWR validation but station BOUL appears not to be an outlier here (for this station, the slope and offset

estimates computed over the full year are closer to normal values). All three versions of the comparisons also reveal a systematic mean bias between GPS and MWR IWV data of about 0.7 kg m$^{-2}$ (0.67 kg m$^{-2}$ with respect to AMSR2), with GPS being drier than MWR. A similar mean bias was previously observed by Mears et al., 2015, on global long-term averages including more GPS sites and satellites. Whether this bias is imbedded in the GPS or MWR retrievals is not clear at the moment. The mean difference between the different satellite estimates is, comparatively, slightly smaller: AMSR2 and Windsat

agree almost perfectly, while GMI has a slight moist bias of +0.2 kg m$^{-2}$ compared to either AMSR2 or Windsat, and F18 has a slight dry bias of –0.4 kg m$^{-2}$ compared to the AMSR2. The slope estimates show more scatter between sites and satellites, although the mean GPS – MWR values agrees very well with the GPS – GPS values. Similar to the findings of Sect. 4.1, the classical correction (v1) does not preserve the slopes and leads to large overestimations for the stations are higher altitudes. Again, the new correction (v2) achieves almost perfect slopes, both for the GPS – GPS and the GPS – MWR comparisons.

Following the IWV inter-comparisons, statistical tests (Appendix C) were applied to sort those comparisons which show biases and offsets significantly different from zero and slopes significantly different from one. Test results with p-values





< 0.01 are highlighted in Fig. 10 for all comparisons. It can be noted that most biases are significant, but not all slopes and offsets. Indeed, the standard errors for the latter parameters remain relatively high, despite one full year of data was used here. For example, several slopes of the GPS – F18 comparisons deviate notably from one, but are not significant (mean standard error of 0.0193), while they are significant for the GPS – GPS intercomparison (mean standard error of 0.0029). These results indicate that not only accurate vertical correction (model v2) and correct specification of the measurement errors are crucial to diagnose biases and scaling errors but also the sample size.

## 5. Discussion and conclusions

In this paper we have shown that the model traditionally used for the correction of the IWV difference due to the vertical displacement between observation sites has two shortcomings. First, it induces a bias in the slope estimate and, correlatively, in the offset estimate, with slopes being over-estimated when the IWV measurements from the lower site are corrected (see, e.g., Fig. 10e). Second, it does not change the offset estimate, which remains generally close to the uncorrected bias value (Figs. 10a, c, and f). We have proposed an improved correction model (Eq. (B8)) based on two terms, $f_c$ and $g_c$, which overcomes these limitations. This model relies on a multi-linear regression of slope and offset (Eq. (B9)) as a function of the height difference (Eqs. (5, 6)). We have shown that high-resolution radiosonde data are capable of providing accurate estimates of the parameters ($a_i, b_i$) of this model on a monthly basis. The correction model reduces the bias, slope, and offset to negligible mean errors (bias $< \pm 0.02$, slope $- 1 < \pm 0.004$, offset $< \pm 0.1$) for height differences up 500 m, with a standard deviation smaller than 0.5 kg m$^{-2}$. The errors are expected to increase slightly for larger height differences (e.g. we found bias $< \pm 0.08$, slope $- 1 < \pm 0.025$, offset $< \pm 0.5$, for a height difference of 1000 m with the data from radiosonde station 78897). The method has been successfully applied to the correction of GPS IWV data from a network of stations in a tropical mountainous area, with altitudes ranging from the sea level up to more than 400 m. Corrected data allowed to diagnose anomalous biases and scaling errors at two sites, which could not be detected in the raw measurements or when the traditional correction method was applied. The method was also applied to inter-validation of IWV from satellite MWR measurements and GPS measurements in the same region. The corrected data confirmed the significant bias and anomalous slope for one of the GPS stations (CBE0, bias close to 2 kg m$^{-2}$, and slope close to 0.98). The reason why the second station (BOUL) did not show up in this comparison is that the errors decreased over time, possibly linked with several equipment changes that were reported during 2020 at this site. Some dispersion was also observed between the four satellite data that were compared, with F18 showing more scatter as well as a smaller number of available collocations. We suspect that the larger footprint of the MWR instrument on board this satellite induced larger representativeness differences, since pixels located farther from the GPS stations have been used. F18 might also have slightly more land contamination than the other satellites do. However, when the results from the four satellites were averaged together, they were in very good agreement with the GPS-only results.

This study also emphasized the need for using a regression method that accounts for errors in both variables and for correctly specifying these errors. Not doing so is known from least-squares theory to result in biased slope and offset estimates,



as well as biased standard errors and inconsistencies in subsequent significance tests. These issues were discussed in Appendix

C, with Monte Carlo simulations, and illustrated in Sect. 4.1, for the case of the GPS – GPS intercomparison. It was namely

shown that the regression method of York et al., 2004, works well as soon as the ratio of the uncertainties in both variables is

properly specified. Stated differently, it appears not necessary to provide absolute uncertainties but only relative ones. This is

fortunate as the former are usually not known, unless an absolute calibration technique is involved. In this study, we have

successfully used a triple collocation method to estimate the relative errors in the GPS and MWR data, using ERA5 as the 3$^{rd}$

data set. This approach provides generally satisfying results as long as the representativeness errors in all data sets are small,

or at least similar (Stoffelen, 1998; O'Carroll et al., 2008). In our case, the MWR and ERA5 have similar spatial resolutions

which may induce representativeness errors of similar magnitude compared to the GPS observations which are of more local

nature. We also attempted to combine GPS, satellite MWR, and radiosonde observations but the triple collocation failed in

this case. However, the combination of collocated GPS, radiosonde, and ground-based MWR measurements from the Barbados

Cloud Observatory during the EUREC4A campaign worked well. In this case, we found the following error estimates: $\sigma_{GPS} =$

0.93 kg m$^{-2}$, $\sigma_{RS} = 0.65$ kg m$^{-2}$, and $\sigma_{MWR} = 1.53$ kg m$^{-2}$. This new estimate for the GPS errors is fairly consistent with the one

found with the satellite MWR and ERA5 data reported in Sect 4.2. It is also consistent with the estimate reported by Cimini et

al., 2012, of 0.94 kg m$^{-2}$. The other two errors seem plausible as well, especially the higher value for the ground-based MWR

data which were shown to contain excessive noise during the first weeks of the campaign (Bock et al., 2021).

430            The improved vertical correction method described in this paper can be easily applied to any other region for which

high resolution vertical profiles of water vapour are available. Such profiles can be provided by radiosonde observations but

also by numerical weather model outputs or by reanalyses. In this study, the model parameters have been derived on a monthly

basis, which seems well adapted to correct data sets which cover at least one month of measurements. We also tested separate

model adjustments and corrections for the 00UTC and 12UTC profiles, but the results we not significantly different. In the

future, we plan to derive the model parameters on a global grid from the ERA5 reanalysis which provides a stable and accurate

climatology of the water vapour distribution. The global correction grid will be useful to provide more accurate inter-

comparisons and inter-validations of global IWV data sets from various techniques.

*Data availability.*

The high-resolution radiosonde data used in this work were retrieved from the University of Wyoming web site

(http://weather.uwyo.edu/upperair/bufrraob.shtml, last access: January 2022). The GPS IWV data are available from AERIS, the French
national data and service portal for the atmosphere (https://www.aeris-data.fr/, last access: January 2022), under DOI
https://doi.org/10.25326/79 (Bock, 2020). The satellite MWR data are freely available via ftp from https://www.remss.com/missions/ after
registration.

**Appendix A: Correction model based on an exponential profile**

The distribution of water vapour in the atmosphere is generally highly variable, but may be approximated by the equation:



$$\rho_v(h) = \rho_0 exp(-\gamma h) , \tag{A1}$$

where $\gamma > 0$ is the mean vertical decay rate of water vapour, also sometimes expressed as the inverse of the water vapour scale height, $H_v = 1/\gamma$, $\rho_0$ is the ground-level water-vapour density, and $h$ is the geometric height. Standard values for $H_v$ and $\rho_0$ are $H_v$= 2 km and $\rho_0$ = 7.5 g m$^{-3}$, or alternatively $\gamma = 5 \cdot 10^{-4}$ m$^{-1}$ (ITU, 2017).

It follows from Eq. (A1) that the IWV above a height $h_A$ is simply:

$$IWV(h_A) = \int_{h_A}^{\infty} \rho_v(h)dh = \frac{\rho_0}{\gamma} exp(-\gamma h_A) , \tag{A2}$$

The IWV in the layer in between two stations, A and B, at heights, $h_A$ and $h_B$, writes:

$$\Delta IWV = \int_{h_A}^{h_B} \rho_v(h)dh = \frac{\rho_0}{\gamma} [\exp(-\gamma h_A) - \exp(-\gamma h_B)], \tag{A3}$$

Equation (A3) can be used to correct the IWV measurements from station A to conform to the height of station B, in an additive

way: $IWV_{A,c} = IWV_A - \Delta IWV$, where $IWV_A = IWV(h_A)$. Combining Eqs. (A2) and (A3) shows that the correction is actually multiplicative in nature:

$$IWV_{A,c} = IWV_A \times \exp(-\gamma(h_B - h_A)) , \tag{A4}$$

Here we can define $f_c(\Delta h)$ as the correction factor which, applied to $IWV_A$, conforms to the height $h_B$:

$$f_c(\Delta h) = \exp(-\gamma \Delta h) \tag{A5}$$

where $\Delta h = h_B - h_A$, is the height difference between station A and station B.

When $|\Delta h|$ is small, or more rigorously when $|\gamma \Delta h| \ll 1$, Eq. (A5) can be approximated by $f_c(\Delta h) \approx 1 - \gamma \Delta h$, which leads to a widely-used form of the IWV correction (Bock et al., 2005; Morland et al., 2006a, b; Buehler et al., 2012):

$$IWV_{A,c} \approx IWV_A - \gamma \cdot \Delta h \cdot IWV_A. \tag{A6}$$

Equation (A6) has traditionally been used to estimate $\gamma$ from the IWV observations at different heights, e.g. for two stations

at heights $h_A$ and $h_B$:

$$\gamma \approx \frac{(IWV_A - IWV_B)}{\Delta h \cdot IWV_A} \tag{A7}$$

which expresses the idea that $\gamma$ represents the fractional IWV variation over a height $\Delta h$ (Bock et al., 2005). The range of validity for the approximate formulations expressed by Eqs. (A6) and (A7) to hold can be estimated from the condition $|\gamma \Delta h| <$ 0.1, which leads to $|\Delta h| < 200$ m, if we use $H_v = \frac{1}{\gamma} = 2$ km. For larger height differences, it is recommended to use the exact

formulations (A4) and (A8):

$$\gamma = -\frac{1}{\Delta h} \log \left( \frac{IWV_B}{IWV_A} \right) \tag{A8}$$



## Appendix B: Link between bias, slope, and offset parameters

Let us assume that we have $n$ observations, $(x_i, y_i)$, $i = 1..n$, corresponding to paired measurements of the same physical quantity coming from the same instrument at two different sites or from two different instruments at the same site. The difference in the observation conditions is assumed to lead a bias, $\Delta$, and a scaling error that can be represented by a linear fit slope, $\alpha$, and offset, $\beta$, defined as:

$$\Delta = \mu_y - \mu_x , \tag{B1}$$

where $\mu_x$ and $\mu_y$ are the sample means of $x$ and $y$, and the slope and offset parameters are derived from the linear regression model:

$$y = \alpha x + \beta, \tag{B2}$$

Thanks to the linearity of the mean operator, the (B2) relationship is also verified for the means:

$$\mu_y = \alpha \mu_x + \beta , \tag{B3}$$

Note that since $\{x_i\}$ and $\{y_i\}$ are both obtained from measurements, they are usually both subject to errors. There exist robust methods to estimate optimally $\alpha$ and $\beta$ in the presence of errors in both variables (e.g. Mandel, 1984; Macdonald and Thompson, 1992; York et al., 2004). Note also that, depending on which of $x$ and $y$ is considered as the reference, the opposite relationship may sometimes be used, $x = \alpha' y + \beta'$, which relates to (B2) by

$$\alpha' = 1/\alpha, \tag{B4a}$$
$$\beta' = -\beta / \alpha. \tag{B4b}$$

Equations (B1) and (B3) recall that the parameters $\Delta$, $\alpha$, and $\beta$ are inter-related through $\mu_x$ and $\mu_y$. It is instructive to discuss the different cases of interest for the interpretation of experimental results. These cases are described below and illustrated in Fig. B1:

- Case n°1: $\alpha = 1$. In this case, the two observation series have only a bias and no scaling error, and it follows from Eqs. (B1) and (B3) that $\mu_y = \mu_x + \beta$ and $\beta = \Delta$.

- Case n°2: $\alpha > 1$. In this case, the two series have a scaling error, where the range of $y_i$ values is larger than the range of $x_i$ values. It also follows from Eqs. (B1) and (B3) that $\beta < \Delta$. Depending on the sign of $\Delta$ there is an additional constraint or not on $\beta$, namely:

    (a) if $\Delta > 0$, then $\beta$ can be either positive or negative, with $\beta < \Delta$.

    (b) if $\Delta < 0$, then $\beta$ can be only negative, i.e., $\beta < \Delta < 0$.

- Case n°3: $\alpha < 1$. In this case, the two series have a scaling error, where the range of $y_i$ values is smaller than the range of $x_i$ values, and it follows from Eqs. (B1) and (B3) that $\beta > \Delta$. Again, there may be an additional constraint on $\beta$:





(a) if $\Delta > 0$, then $\beta$ can be only positive: $\beta > \Delta > 0$.

(b) if $\Delta < 0$, then $\beta$ can be either positive or negative, with $\beta > \Delta$.

Let us now analyse the impact of applying a rescaling of the reference series, $\{x_i\}$, in order to correct it for difference in the observation conditions with respect to the tested series $\{y_i\}$. We denote the corrected series by $\{x_{i,c}\}$, with $x_{i,c} = f_c \times x_i$. In the case of IWV vertical correction, the scaling factor $f_c$ could be computed from Eq. (A5) under the hypothesis of a vertical distribution of water vapour following an exponential law. The bias, slope, and offset parameters, after correction, are denoted respectively as $\Delta_c$, $\alpha_c$, and $\beta_c$, and write:

$$\Delta_c = \mu_y - f_c \mu_x \,, \tag{B5}$$

$$\alpha_c = \frac{\alpha}{f_c}\,, \tag{B6}$$

$$\beta_c = \beta\,, \tag{B7}$$

Equations (B5) and (B6) follow from the fact that $\mu_y = \alpha \mu_x + \beta = \alpha_c f_c \mu_x + \beta_c$ must hold for every $\mu_x$. A crucial question is to check if this correction method can achieve a perfect bias correction and scaling simultaneously, i.e. $\Delta_c = 0$ and $\alpha_c = 1$.

Let us first check the conditions for achieving a zero bias, $\Delta_c = 0$. This result is achieved if and only if $f_c = \mu_y / \mu_x$. From this condition, it follows that $\alpha_c = \alpha \frac{\Delta - \beta}{\Delta \alpha - \beta}$. The only possibility to simultaneously achieve $\Delta_c = 0$ and $\alpha_c = 1$ is actually that $\alpha = 1$, i.e. when the data have initially only a bias but no scaling error. In all other cases, the final slope will be different from one, and it can be either larger or smaller than the initial slope, i.e. in some cases, the slope can be degraded (getting farther from one). These situations depend again on the initial values of $\alpha$ and $\Delta$:

- case n°1: if $\Delta > 0$, then $\alpha_c < \alpha$. If, in addition, $\alpha < 1$ then $\alpha_c < \alpha < 1$, i.e. the slope is degraded. If, instead, $\alpha > 1$, then $\alpha_c < \alpha$ can lead to an improvement in the slope but there is no guarantee that $\alpha_c$ will be close to one.

- case n°2: if $\Delta < 0$, then $\alpha_c > \alpha$. If, in addition, $\alpha > 1$ then $\alpha_c > \alpha > 1$, i.e. the slope is degraded. If, instead, $\alpha < 1$, then $\alpha_c > \alpha$ can lead to an improvement in the slope but there is no guarantee that $\alpha_c$ will be close to one.

The above analysis shows that, except when $\alpha = 1$, the final slope will be different from one, and in some cases, depending
on the sign of the initial bias, it will be degraded.

Let us now check the conditions for achieving a unity slope, $\alpha_c = 1$. This result is achieved if and only if $f_c = \alpha$. From there it results that $\Delta_c = \beta$, i.e. the sign and magnitude of the final bias will depend on the sign and magnitude of the initial offset. Unless $\beta = 0$, the final bias will generally be different from zero, i.e., it is in general not possible to achieve a zero bias if the reference data are corrected by a simple scaling factor that would achieve a final slope of one.





530        Instead of a simple rescaling correction model, we propose to use a linear correction model which includes both a scaling factor, $f_c$, and an intercept, $g_c$:

$$x_{i,c} = f_c \times x_i + g_c \qquad (B8)$$

For our application to IWV vertical correction, both $f_c$ and $g_c$ would depend on the height difference, $\Delta h$. Following the same reasoning as for the simple scaling model, it is straightforward to show that the condition to achieve both a zero bias, $\Delta_c = 0$,

and an unity slope, $\alpha_c = 1$, after correction writes:

$$f_c = \alpha \text{ and } g_c = \beta \qquad (B9)$$

Indeed, substituting (B9) into (B8), and expressing the bias $\Delta_c = \mu_y - \mu_{x,c}$, and the linear fit equation $\mu_y = \alpha_c \mu_{x,c} + \beta_c$, after correction, we find $\Delta_c = 0$, $\alpha_c = 1$, and $\beta_c = 0$, which is the desired result.

**Appendix C: Statistical properties of the bias and straight line fitted parameters**

The classical straight line fitting problem can be formalized as follows. Let us assume the linear model

$$Y = \alpha x + \beta + \varepsilon_Y, \qquad (C1)$$

where $Y$ is the response variable, $x$ the independent variable, $\alpha$ and $\beta$ the slope and intercept, and $\varepsilon_Y$ a random variable of zero mean and variance $\sigma_{\varepsilon,Y}^2$, representing the error in $Y$. When $x$ is known without error, the ordinary least squares (OLS) solution is found by minimizing the sum of squared errors, $SSE = \sum_{i=1}^n e_i^2$, where $e_i = y_i - \hat{y}_i$ and $\hat{y}_i = \hat{\alpha} x_i + \hat{\beta}$ is the predicted value from the fitted line. In this case, the errors represent the vertical distance of the best fit line to the data points. The OLS solution

for $\alpha$ and $\beta$ has a simple analytical formulation (Walpole, 2012):

$$\alpha_{OLS} = \frac{\sum_{i=1}^n (x_i - \bar{x})(y_i - \bar{y})}{\sum_{i=1}^n (x_i - \bar{x})^2}, \qquad (C2a)$$

$$\beta_{OLS} = \bar{y} - \alpha_{OLS} \bar{x}. \qquad (C2b)$$

The variance of the OLS estimators is given by (Walpole, 2012):

$$\sigma_{\alpha,OLS}^2 = \frac{1}{\sum_{i=1}^n (x_i - \bar{x})^2} \sigma_{\varepsilon,Y}^2, \qquad (C3a)$$

$$\sigma_{\beta,OLS}^2 = \frac{\sum_{i=1}^n x_i^2}{\sum_{i=1}^n (x_i - \bar{x})^2} \sigma_{\varepsilon,Y}^2. \qquad (C3b)$$

An unbiased estimate of $\sigma_{\varepsilon,Y}^2$ is given by (Walpole, 2012):

$$s_{\varepsilon,Y}^2 = \frac{SSE}{n-2} = \sum_{i=1}^n \frac{(y_i - \hat{y}_i)^2}{n-2} \qquad (C4)$$





From there, it is customary to compute the standard errors of the estimates as:

$$se_{\alpha,OLS}^2 = \frac{1}{\sum_{i=1}^n (x_i - \bar{x})^2} s_{\varepsilon,Y}^2, \tag{C5a}$$

$$se_{\beta,OLS}^2 = \frac{\sum_{i=1}^n x_i^2}{\sum_{i=1}^n (x_i - \bar{x})^2} s_{\varepsilon,Y}^2. \tag{C5b}$$

Assuming that the errors $\varepsilon_{Y,i}$ are normally distributed, it follows that the estimators $\alpha_{OLS}$ and $\beta_{OLS}$ are also normally distributed, and that $(n-2)s_{\varepsilon,Y}^2/\sigma_{\varepsilon,Y}^2$ is a chi-squared variable with $n-2$ degrees of freedom. Hypothesis testing of the fitted parameters is then done using the following statistics:

$$t_{\alpha,OLS} = \frac{\alpha_{OLS} - \alpha_0}{se_{\alpha,OLS}} \tag{C6a}$$

$$t_{\beta,OLS} = \frac{\beta_{OLS} - \beta_0}{se_{\beta,OLS}} \tag{C6b}$$

which both have $t$-distributions with $n-2$ degrees of freedom. In (C6a) and (C6b), $\alpha_0$ and $\beta_0$ are the values assumed in the null hypotheses. Typically, one wants to test $H_0$: $\alpha_{OLS} = 1$ against $H_1$: $\alpha_{OLS} \neq 1$, and $H_0$: $\beta_{OLS} = 0$ against $H_1$: $\beta_{OLS} \neq 0$. The associated p-values are then computed from the $t$-cumulative distribution function (CDF):

$$p_{\alpha,OLS} = 2 \cdot tcdf(-|t_{\alpha,OLS}|, n-2) \tag{C7a}$$

$$p_{\beta,OLS} = 2 \cdot tcdf(-|t_{\beta,OLS}|, n-2) \tag{C7b}$$

When $x$ is observed with error, a second observing equation applies:

$$X = x + \varepsilon_X, \tag{C8}$$

where the observed quantity $X$ of the unknown variable $x$, now contains a random error $\varepsilon_X$, and the OLS solution (C2) is no longer optimal. Indeed, the slope estimate will typically have negative bias (see Draper and Smith, 1998, Eq. (3.4.10) for an expression of the bias) and this will bias the intercept estimate in return.

The solution of the regression of $Y$ on $X$ with errors in both variables, can be found by minimizing the sum of squared errors in both variables: $SSE = \sum_{i=1}^n [w(x_i)(x_i - \hat{x}_i)^2 + w(y_i)(y_i - \hat{y}_i)^2]$ where $w(x_i)$ and $w(y_i)$ are the weights of the observations, and $\hat{x}_i$ and $\hat{y}_i$ are the predicted values. Weights have been included here to follow the formalism of York et al., 2004. They would typically be computed from the assumed uncertainties, $u$, in the measurements, e.g. $w(x_i) = 1/u_{i,x}^2$ and $w(y_i) = 1/u_{i,y}^2$. Note that in the special case of unit weights, the solution is the straight line that minimizes the sum of the squares of the perpendicular distances to the observed points (Macdonald and Thompson, 1992). Finding the solution of this problem is not straightforward and many different, often approximate, solutions have been proposed in the literature (see e.g. the discussion in Press et al., 2007). In this work, we use the iterative algorithm approach proposed by York et al., 2004, which includes also equations for the standard errors of the fitted parameters. The equations are more complex than those of the OLS



solution and will not be repeated here. The standard errors of the York estimators can be likewise used with Eqs. (C6-7) for hypothesis testing. However, here we want to emphasize that the formulations of the standard errors given by York et al., 2004, need to be rescaled by the goodness of fit factor $\sqrt{SSE/(n-2)}$, where $SSE$ is the residual sum of squares given above. This rescaling is important to retrieve realistic values of the standard errors and thus the test statistics and the subsequent p-values.

If the uncertainties in the measurements have been properly specified, this quantity should be close to 1.

In addition, it is useful to describe how the bias estimates can be tested. Especially, we want to test the null hypothesis: $H_0: \Delta = 0$ against $H_1: \Delta \neq 0$, where $\Delta$ is computed as $\Delta = \bar{y} - \bar{x}$. The difficulty here is with standard error of $\Delta$ when both variables have errors. It can be shown that the mean and variance of the $\Delta$ estimator are: $E(\Delta) = (\alpha - 1)\bar{x} + \beta$ and $Var(\Delta) = \sigma_\Delta^2 = \frac{\sigma_{\varepsilon,X}^2 + \sigma_{\varepsilon,Y}^2}{n}$, respectively. The problem here is that $\sigma_{\varepsilon,X}^2$ and $\sigma_{\varepsilon,Y}^2$ are unknown. It may be conjectured that $s_\delta^2 = \sum_{i=1}^n \frac{(\delta_i - \bar{\delta})^2}{n-1}$

is a proper estimator of the variance of $Y - X$, with $\delta_i = y_i - x_i$ and $\bar{\delta} = \bar{y} - \bar{x}$, and that $\frac{s_\delta^2}{n}$ may be used as an estimator of $\sigma_\Delta^2$. However, it can be shown that $E(s_\delta^2) = (1 - \alpha)^2 \sum_{i=1}^n \frac{(x_i - \bar{x})^2}{n-1} + \sigma_{\varepsilon,X}^2 + \sigma_{\varepsilon,Y}^2$ and that the first term is typically dominant over the latter two, i.e. this estimator of $\sigma_\Delta^2$ is biased. Instead, we propose to use (C4) as an estimator of $\sigma_{\varepsilon,Y}^2$ and a similar estimator for $\sigma_{\varepsilon,X}^2$:

$$s_{\varepsilon,X}^2 = \sum_{i=1}^n \frac{(x_i - \hat{x}_i)^2}{n-2} \qquad \text{(C9)}$$

where $\hat{x}_i = \frac{y_i - \hat{\beta}}{\hat{\alpha}}$ is the predicted value for $x_i$. It can be shown that $E(s_{\varepsilon,Y}^2) \approx \alpha^2 \sigma_{\varepsilon,X}^2 + \sigma_{\varepsilon,Y}^2$ and $E(s_{\varepsilon,X}^2) \approx \sigma_{\varepsilon,X}^2 + \frac{\sigma_{\varepsilon,Y}^2}{\alpha^2}$. Since in our applications, $\alpha$ is usually close to one, and the two error variances are comparable, $\sigma_{\varepsilon,X}^2 \approx \sigma_{\varepsilon,Y}^2$, both estimators will only depart slightly from $\sigma_{\varepsilon,X}^2 + \sigma_{\varepsilon,Y}^2$. In consequence, we propose to average the two estimates, and use

$$s_\Delta^2 = \frac{s_{\varepsilon,X}^2 + s_{\varepsilon,Y}^2}{2n} \qquad \text{(C10)}$$

as an estimator of $\sigma_\Delta^2 = Var(\Delta)$. Note that the OLS and York estimators predict different values of $s_\Delta^2$ because $s_{\varepsilon,X}^2$ and $s_{\varepsilon,Y}^2$

depend on the estimated values of $\alpha$ and $\beta$. The test statistic and subsequent p-value for $\Delta$ can be computed in a similar manner as for $\alpha$ and $\beta$, although the statistic does not exactly follow a $t$-distribution in this case.

The performance of the OLS and York regression methods have been evaluated based on Monte Carlo tests. The main goals were to evaluate: i) the impact of errors in $x$ on the OLS estimator, ii) the impact of mis-specification of the errors in the two variables with the York estimator, iii) the performance of the test statistic for the bias. We simulated $m = 10^5$ data sets,

each composed of $n = 41$ pairs of observations, $(x_i, y_i)$, $i = 1..n$, where $x_i = 10..50$ by step of 1 plus a random value from a normal distribution, $N(0, \sigma_X^2)$, and $y_i = \alpha \tilde{x}_i + \beta$ plus a random value from a normal distribution, $N(0, \sigma_Y^2)$, where $\tilde{x}_i$ is the true (noise-free) value of $x_i$. Table C1 presents the results for different cases where the true noise variances, $\sigma_X^2$ and $\sigma_Y^2$, were changed, and the assumed variances, $u_x^2$ and $u_y^2$, were either correctly specified or not (note that the latter are used only in the





York fit). All these simulations were run with $\alpha = 1$ and $\beta = 0$. We also run simulations for other values of $\alpha$ and $\beta$, but the

conclusions were unchanged, e.g., with $\alpha = 0.8$ and $\beta = 5.0$ we did not observe any significant difference in the results compared to those presented in Table C1. Note that the SE values for $\Delta$ reported in Table C1 were computed with the York estimates of $\alpha$ and $\beta$. We observed that they were consistent with the values computed with the OLS estimates to 0.01 or better (OLS values greater than York values) and consequently led to the same hypothesis test results on average.

      The performance of the estimators was assessed in terms of bias (difference between the mean estimate and the truth),

variance (the consistency between the observed standard deviation, STD, and the mean standard error, SE), and the correctness of the 5% significance level (the value $p_{0.05}$ reported in Table C1 is the fraction of simulations with p-values $< 0.05$). The results are summarized below:

- $\sigma_X^2 = 0$, when no errors are simulated in $x$ (case n°1), the OLS and York methods yield identical results (mean, STD, SE, $p_{0.05}$)


- $\sigma_X^2 > 0$, the OLS estimates of slope and offset are biased ($\alpha_{OLS} < 1$ and $\beta_{OLS} > 0$) and the magnitudes of the biases depend on the strength of the noise:
  - when $\sigma_X = 1$ (cases n°2, 3, 4, 6, 8), the biases are small: $\alpha_{OLS} \approx 0.993$, $\beta_{OLS} \approx 0.19$, and $p_{0.05} \approx 0.06$,
  - when $\sigma_X = 4$ (cases n°5, 7), the biases are larger: $\alpha_{OLS} \approx 0.90$, $\beta_{OLS} \approx 2.9$, and $p_{0.05} \approx 0.5$,

  - when $\sigma_X$ is proportional to X (cases n°9, 10), the biases take intermediate values: $\alpha_{OLS} = 0.93 - 0.98$, $\beta_{OLS} = 0.5 - 1.9$, and $p_{0.05} = 0.1 - 0.2$,
  - when $\sigma_Y > \sigma_X$ (cases n°6, 8), the mean values are unchanged, but STD and SE increase, and $p_{0.05}$ is improved (compare, e.g., cases n°2 and 6).

- $\sigma_X^2 > 0$, the York estimates are unbiased in all cases, except when the uncertainties are mis-specified and

they are dissimilar in both variables:
  - when $\frac{u_X}{u_Y} \neq \frac{\sigma_X}{\sigma_Y}$ (cases n°3, 5, 6), the biases amount to $\alpha_{York} - 1 = \pm 0.05$, $\beta_{York} = \pm 1.5$, $p_{0.05} = 0.14 - 0.18$ in cases n°5, 6, but they are much smaller in case n°3.
  - when $\frac{u_X}{u_Y} = \frac{\sigma_X}{\sigma_Y}$ (cases n°7, 8), all the biases vanish, although the specified uncertainties are smaller than the true errors.


- The standard errors are consistent with the standard deviations in all cases. They increase when the noise increases. Note that the standard errors are relatively large in these simulations because the samples contain only $n = 41$ values. Increasing $n$ by a factor of 10 (consistent with the GPS – MWR comparisons of Sect. 4.2), would decrease SE by a factor of 10. A reduction of the SE would also imply that some of the slope and offset biases become significant (e.g. in case n°3).



- The bias estimator, $\Delta = \bar{y} - \bar{x}$, is "unbiased" in all cases (mean ≈ true value) and its SE estimator is consistent with the standard deviation (this confirms the validity of the SE estimator given by Eq. (C10)) with only a small bias when the noise variances are dissimilar (cases 5, 6: SE ≈ 0.62 compared to STD = 0.64) and subsequent impact on the $p_{0.05}$ probabilities ($p_{0.05}$ = 0.13 - 0.18).

Note that even when $\sigma_X^2 > 0$ is constant, the OLS estimators and subsequent test statistics are biased.

Figure C1 shows the distributions of the slope, offset and p-values from the hypothesis tests for cases n°5 and 7. The shapes of the distributions of the slope and offset resemble non-central *t*-distributions. Note the biases of the OLS estimators in both cases and the bias in the York estimators only in case n°5. The distributions of the p-values are expected to be flat (equal probability for all p-values), which is verified for the York fit in case n°7, and all other simulated cases, except for cases n°5 and 6, and to a lesser extent case n°3, when the error ratios are mis-specified. In case n°5 shown in the figure, it is seen

that the small p-values have larger probability, which indicates that an excessive number of slope and offset estimates are biased. This happens more often with the OLS estimator.

*Author contributions.* OB initiated the study, proposed the new correction method, performed the comparisons, and wrote the paper. PB and CM processed the GPS data and MWR data used in this work, respectively. All three authors contributed to the interpretation and discussion of the results.

*Competing interests.* The authors declare that they have no conflict of interest.

*Special issue statement.* This article is part of the special issue "Analysis of atmospheric water vapour observations and their uncertainties for climate applications (ACP/AMT/ESSD/HESS inter-journal SI)". It is not associated with a conference.

*Acknowledgments.* The authors would like to thank Guillaume Gamelin (Meteo-France), for providing assistance in the quality checking of the high resolution radiosonde data of station Le Raizet, Guadeloupe, France. The authors are grateful to AERIS (https://www.aeris-
data.fr/), the French data and service centre for atmosphere, for providing the ERA5 reanalysis data. This work was developed in the framework of the VEGAN project supported by the CNRS program LEFE/INSU.

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



Table 1: Height above sea level and number of IWV estimates (N) for 15 GNSS stations over the Guadeloupe islands (62°W-61°W, 15.75°N-16.75°N), for the period from 1 January to 29 February 2020 (Bock et al. 2021).

|  | PPTG | LDIS | DEHA | DESI | MAGT | BOUL | ABMF | ABD0 | ABER | GOSI | FFE0 | MAGA | FNA0 | CBE0 | HOUE |
|---|---|---|---|---|---|---|---|---|---|---|---|---|---|---|---|
| Height (m) | 1 | 4 | 5 | 11 | 13 | 14 | 15 | 20 | 25 | 49 | 53 | 62 | 122 | 374 | 418 |
| N | 1281 | 1439 | 1423 | 1208 | 1439 | 1199 | 1438 | 1380 | 1209 | 1165 | 1439 | 1191 | 474 | 752 | 1429 |



Table 2: Median values and smallest absolute values for the bias, slope (or slope – 1), and offset of GNSS IWV inter-comparisons after correction for the height difference using the proposed model (v2). Data cover the period from 1 January to 29 February 2020. Slope and offset are estimated with the York et al. 2004, method. Bias and offset values significantly different from 0, and slope values significantly different from 1, are highlighted depending on their p-value (* ≤ 0.05, ** ≤ 0.01). Two anomalous stations (BOUL and CBE0) with large bias and offset values, and slope deviating from one, are

highlighted in bold.

| | PPTG | LDIS | DEHA | DESI | MAGT | BOUL | ABMF | ABD0 | ABER | GOSI | FFE0 | MAGA | FNA0 | CBE0 | HOUE |
|---|---|---|---|---|---|---|---|---|---|---|---|---|---|---|---|
| Median value | | | | | | | | | | | | | | | |
| Bias (kg m$^{-2}$) | -0.21** | -0.41** | 0.51** | -0.53** | 0.03 | -0.04 | 0.29** | -0.42** | -0.12 | 0.21** | 0.50** | 0.11 | 0.05 | **2.01**\** | -0.31** |
| Slope | 1.011* | 0.965** | 1.021* | 0.970* | 0.993 | **1.120**\** | 1.011 | 0.984 | 0.999* | 0.997 | 1.012 | 1.020 | 0.983 | **0.940**\** | 1.008 |
| Offset (kg m$^{-2}$) | -0.57 | 0.70 | -0.10 | 0.46 | 0.05 | **-4.09**\** | -0.16 | -0.10 | -0.10 | 0.18 | 0.10 | -0.77 | 0.64 | **3.73**\** | -0.48 |
| Smallest absolute value | | | | | | | | | | | | | | | |
| Bias (kg m$^{-2}$) | 0.05 | 0.02 | 0.01 | 0.08 | 0.01 | 0.01 | 0.04 | 0.02 | 0.09 | 0.04 | 0.01 | 0.09** | 0.07 | **1.56**\** | 0.05 |
| Slope – 1 | 0.001 | 0.004* | 0.002 | 0.004* | 0.002 | **0.065**\** | 0.003 | 0.002 | 0.006* | 0.000 | 0.001 | 0.004 | 0.003 | **0.027*** | 0.000 |
| Offset (kg m$^{-2}$) | 0.11 | 0.04 | 0.00 | 0.04 | 0.03 | **2.33**\** | 0.11 | 0.00 | 0.10 | 0.07 | 0.06 | 0.07 | 0.15 | **2.57**\** | 0.11 |





Table 3: Mean distance between GPS stations and the nearest MWR satellite grid-point within the 7 x 7 box surrounding each
station (MWR grid resolution 0.25°x0.25°). Units: km. NA=not available.

|  | PPTG | LDIS | DEHA | DESI | MAGT | BOUL | ABMF | ABD0 | ABER | GOSI | FFE0 | MAGA | FNA0 | CBE0 | HOUE |
|---|---|---|---|---|---|---|---|---|---|---|---|---|---|---|---|
| AMSR2 | 40.0 | 20.0 | 20.7 | 20.0 | 46.3 | 26.3 | 20.5 | 42.4 | 41.5 | 45.5 | 42.1 | 45.7 | 30.8 | 41.1 | 21.0 |
| F18 | 80.1 | NA | 73.8 | NA | 77.1 | NA | 81.9 | 93.5 | 93.8 | 82.5 | 82.2 | 76.3 | NA | 82.0 | NA |
| GMI | 38.8 | 20.5 | 11.7 | 20.5 | 21.2 | 12.7 | 18.5 | 40.5 | 40.3 | 42.5 | 40.5 | 20.4 | 28.1 | 29.7 | 18.3 |
| Windsat | 40.8 | 20.5 | 22.3 | 20.5 | 47.3 | 29.2 | 40.2 | 54.8 | 54.8 | 46.6 | 43.0 | 46.5 | 31.5 | 44.5 | 21.7 |





Table C1: Monte Carlo tests of linear regression using the ordinary least-squares (OLS) and the York et al. (2004) method. In all simulated cases, the true slope is 1 and true offset is 0. Noise is simulated in both variables according to the standard deviation values indicated in the first two columns. The OLS method assumes noise is present only in the *y* variable. The York method accounts for errors in both variables, with uncertainties specified in the 3rd and 4th columns. The other columns report the mean and standard deviation (STD) of estimated parameters (bias, slope, and offset), and their mean standard errors (SE). The column "p<0.05" indicates the fraction of results that have p-values smaller than 0.05. The expected value for the latter is 0.05. Each data set was run for $10^5$ simulations. Mean values and "p<0.05" values which differ significantly from the expected values are highlighted in bold.

| | Simulated noise | | Assumed noise | | OLS | | | | | | | | | | | |
| | | | | | Bias | | | | Slope | | | | Offset | | | |
| | $\sigma_x$ | $\sigma_y$ | $u_x$ | $u_y$ | mean | STD | SE | p<0.05 | mean | STD | SE | p<0.05 | mean | STD | SE | p<0.05 |
|---|---|---|---|---|---|---|---|---|---|---|---|---|---|---|---|---|
| 1 | 0 | 1 | 0 | 1 | -0.0005 | 0.1561 | 0.1533 | 0.0507 | 1.0000 | 0.0132 | 0.0131 | 0.0495 | -0.0011 | 0.4272 | 0.4231 | 0.0501 |
| 2 | 1 | 1 | 1 | 1 | 0.0001 | 0.2207 | 0.2165 | 0.0500 | 0.9934 | 0.0186 | 0.0184 | **0.0659** | 0.1991 | 0.5984 | 0.5950 | **0.0636** |
| 3 | 1 | 1 | 4 | 1 | 0.0015 | 0.2203 | 0.2170 | 0.0574 | 0.9935 | 0.0185 | 0.0185 | **0.0644** | 0.1964 | 0.5973 | 0.5954 | **0.0619** |
| 4 | 1 | 1 | 4 | 4 | 0.0000 | 0.2207 | 0.2167 | 0.0513 | 0.9934 | 0.0185 | 0.0185 | **0.0656** | 0.1983 | 0.5980 | 0.5955 | **0.0633** |
| 5 | 4 | 1 | 1 | 1 | 0.0013 | 0.6445 | 0.6261 | **0.1833** | **0.9038** | 0.0453 | 0.0491 | **0.4922** | **2.8825** | 1.4823 | 1.5955 | **0.4343** |
| 6 | 1 | 4 | 1 | 1 | -0.0002 | 0.6450 | 0.6261 | **0.1358** | 0.9937 | 0.0545 | 0.0539 | 0.0515 | 0.1890 | 1.7580 | 1.7385 | 0.0516 |
| 7 | 4 | 1 | 1 | 0.25 | 0.0031 | 0.6431 | 0.6326 | 0.0508 | **0.9038** | 0.0453 | 0.0491 | **0.4933** | **2.8833** | 1.4820 | 1.5955 | **0.4330** |
| 8 | 1 | 4 | 0.25 | 1 | -0.0026 | 0.6441 | 0.6325 | 0.0525 | 0.9937 | 0.0543 | 0.0539 | 0.0517 | 0.1870 | 1.7503 | 1.7379 | 0.0507 |
| 9 | 5% | 5% | 5% | 5% | -0.0007 | 0.3567 | 0.3499 | 0.0512 | 0.9831 | 0.0311 | 0.0294 | **0.1021** | 0.5087 | 0.7585 | 0.9490 | **0.0370** |
| 10 | 10% | 10% | 10% | 10% | 0.0001 | 0.7130 | 0.7005 | 0.0549 | **0.9359** | 0.0597 | 0.0567 | **0.2175** | **1.9362** | 1.4757 | 1.8371 | **0.1256** |

| | Simulated noise | | Assumed noise | | York fit | | | | | | | |
| | | | | | Slope | | | | Offset | | | |
| | $\sigma_x$ | $\sigma_y$ | $u_x$ | $u_y$ | mean | STD | SE | p<0.05 | mean | STD | SE) | p<0.05 |
|---|---|---|---|---|---|---|---|---|---|---|---|---|
| 1 | 0 | 1 | 0 | 1 | 1.0000 | 0.0132 | 0.0131 | 0.0495 | -0.0011 | 0.4272 | 0.4231 | 0.0501 |
| 2 | 1 | 1 | 1 | 1 | 1.0001 | 0.0188 | 0.0185 | 0.0514 | -0.0032 | 0.6040 | 0.5969 | 0.0517 |
| 3 | 1 | 1 | 4 | 1 | **1.0063** | 0.0188 | 0.0187 | **0.0576** | **-0.1860** | 0.6045 | 0.6022 | **0.0560** |
| 4 | 1 | 1 | 4 | 4 | 1.0001 | 0.0187 | 0.0185 | 0.0511 | -0.0044 | 0.6038 | 0.5974 | 0.0508 |
| 5 | 4 | 1 | 1 | 1 | **0.9530** | 0.0513 | 0.0504 | **0.1805** | **1.4073** | 1.6586 | 1.6340 | **0.1620** |
| 6 | 1 | 4 | 1 | 1 | **1.0527** | 0.0565 | 0.0556 | **0.1404** | **-1.5818** | 1.8132 | 1.7890 | **0.1269** |
| 7 | 4 | 1 | 1 | 0.25 | 1.0029 | 0.0552 | 0.0543 | 0.0516 | -0.0906 | 1.7783 | 1.7518 | 0.0511 |
| 8 | 1 | 4 | 0.25 | 1 | 1.0004 | 0.0546 | 0.0539 | 0.0516 | -0.0155 | 1.7611 | 1.7386 | 0.0507 |
| 9 | 5% | 5% | 5% | 5% | 1.0002 | 0.0245 | 0.0243 | 0.0498 | -0.0047 | 0.5383 | 0.5326 | 0.0508 |
| 10 | 10% | 10% | 10% | 10% | 1.0014 | 0.0498 | 0.0483 | 0.0543 | -0.0298 | 1.0897 | 1.0605 | 0.0541 |





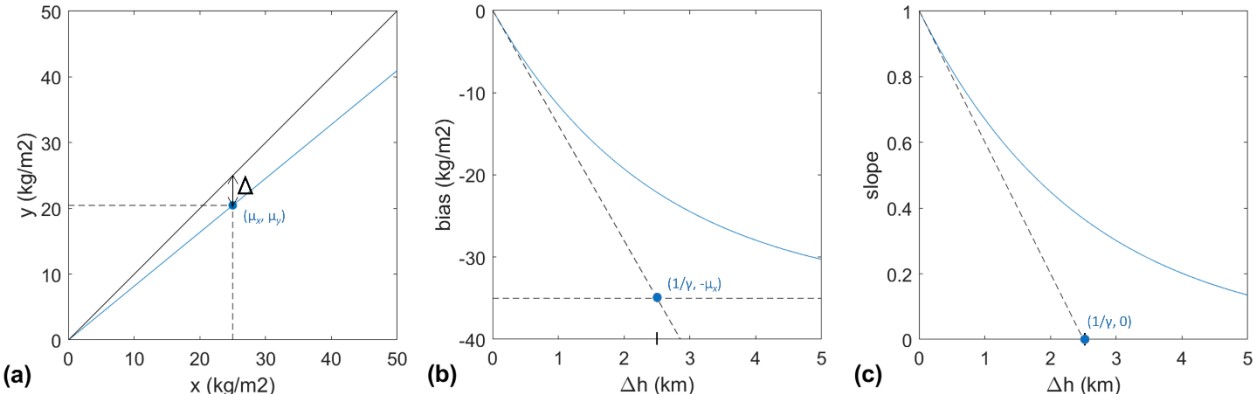

Figure 1: Illustration of the variation of IWV as a function of height in the case of an idealized moisture profile with exponential vertical decay with a rate $\gamma = 4 \cdot 10^{-4}$ m$^{-1}$ (scale height $1/\gamma = 2.5$ km); (a) $y = x \cdot \exp(-\gamma \Delta h)$ as a function of $x$ for a fixed $\Delta h > 0$; (b) bias, $\Delta = \mu_y - \mu_x$, as a function of $\Delta h$ (Eq. (3)); (c) slope, $\alpha$, as a function of $\Delta h$ (Eq. (4)). The slant dashed lines in (b) and (c) represent the thin layer approximations (last right-hand sides of Eqs. (3) and (4), respectively).





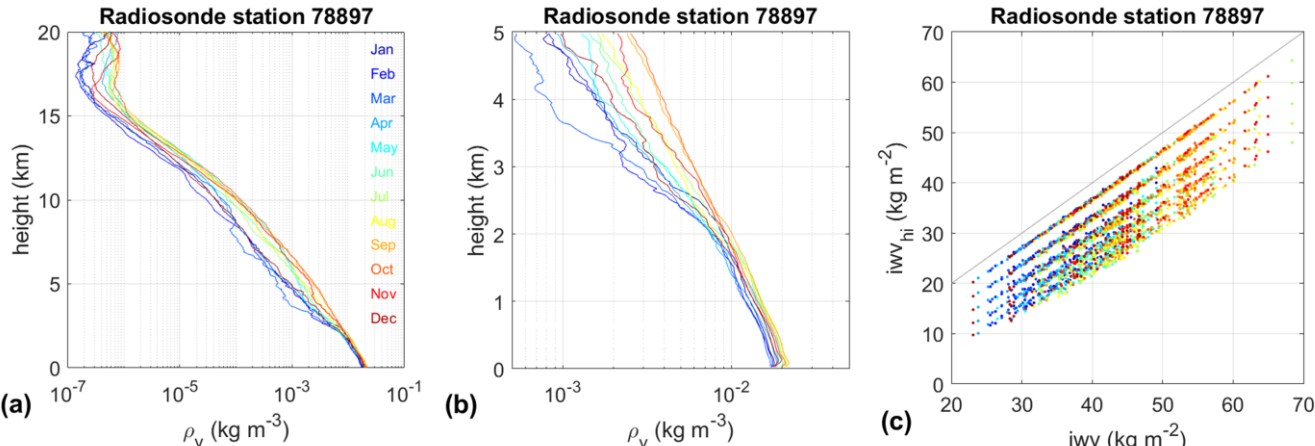

Figure 2: Real water vapour profiles observed by radiosonde station 78897 (Le Raizet, Guadeloupe, France): (a) monthly mean profiles for year 2020; (b) similar to (a) for altitudes below 5 km; (c) IWV scatter plot, with upper-level IWV plotted on the $y$-axis, and total column IWV on the $x$-axis, for five different height differences, $\Delta h$=200, 400, 600, 800, 1000 m. The radiosonde data include 00UTC and 12UTC soundings. The colour code indicated in (a) is valid for all plots.




Figure 3: Monthly mean estimates computed from the radiosonde observations shown in Fig. 2c, for $\Delta h$=25 to 1000 m, by step of 25 m: (a, d) bias, $\Delta$, and relative bias, $\Delta/\mu_x$; (b, c) slope and offset parameters fitted from Eq. (B2) by ordinary least-squares; (e, f) standard errors of the slope and offset parameters.






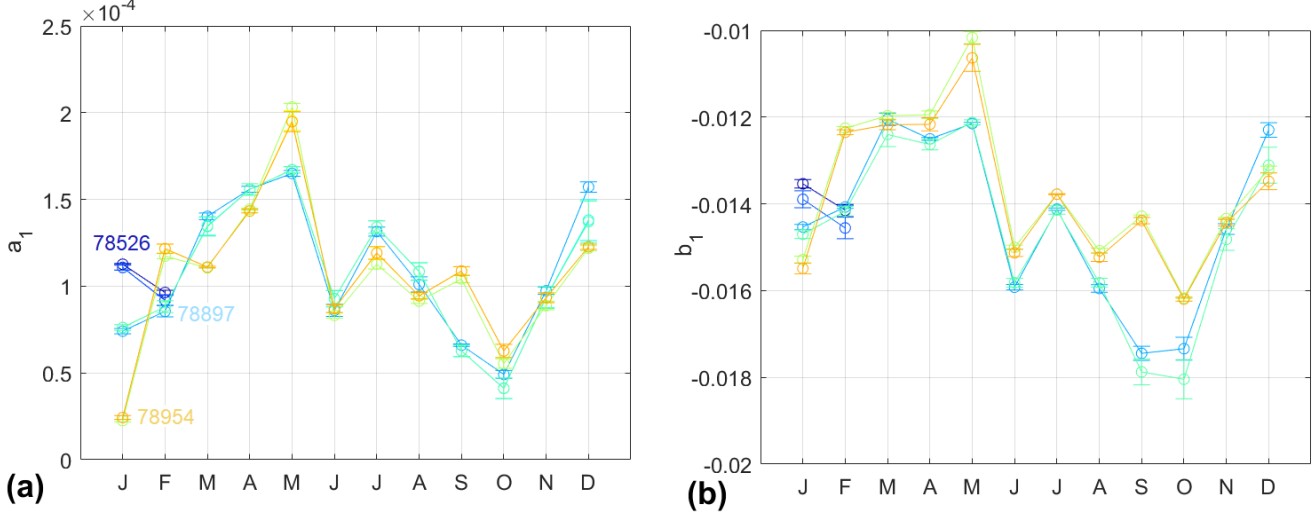

Figure 4: Monthly estimates of the polynomial coefficients for the slope (a) and the offset (b), according to Eqs. (6a) and (6b),

respectively, limited to order 1. The different curves show results for three different radiosonde stations (labelled by their WMO codes: 78897, 78954, and 78526) and two regression methods (OLS and WLS). The OLS and WLS results are almost superposed and are not labelled. Note that for station 78526 only two months of observations (January and February) were available in 2020. The error bars indicate the 95% confidence interval for each estimated parameter.




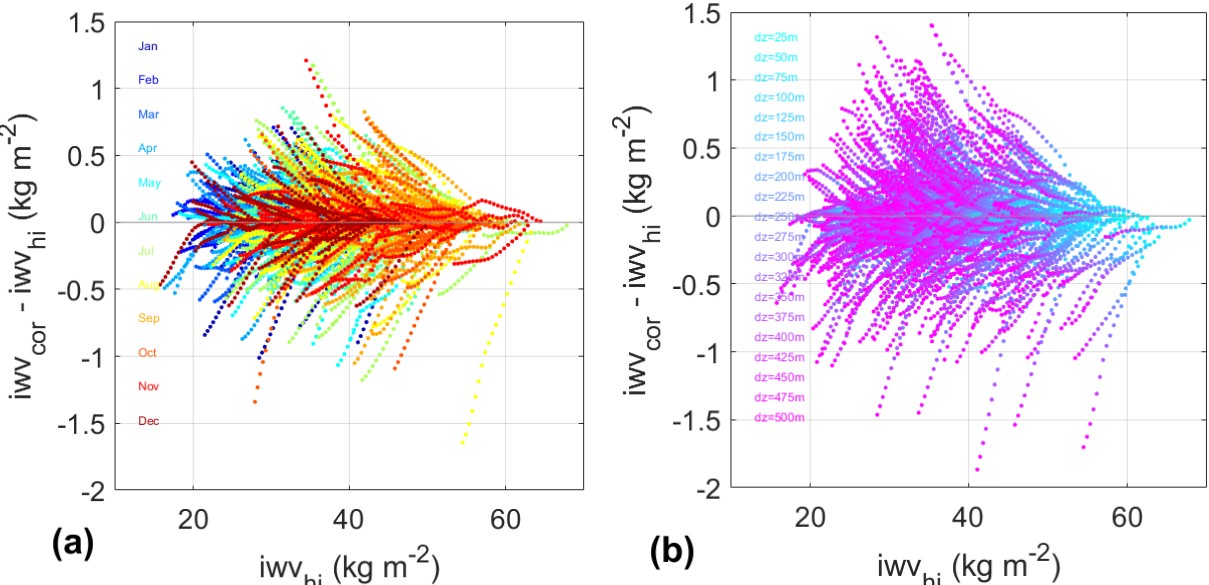

Figure 5: IWV correction error, $x_{i,c} - y_i$, with: (a) monthly coefficients, and (b) yearly coefficients, as a function of time and height difference, $\Delta h$=25..500 m. Both models used polynomials of order 5 and weighted least-squares estimation. The time is colour-coded in (a), while the height difference is colour coded in (b). The dots aligned in filaments correspond to a given time and varying $\Delta h$, in both plots.

Figure 6: Monthly mean bias (a) and standard deviation of the IWV correction error (b) with the monthly coefficients up to order 5 and WLS, and slope (c) and offset (d) parameters, $\alpha_c$ and $\beta_c$, of the best linear fit after correction, $y_i = \alpha_c x_{i,c} + \beta_c$, as a function of time and height difference. The colour code for time is the same as in Fig. 2.



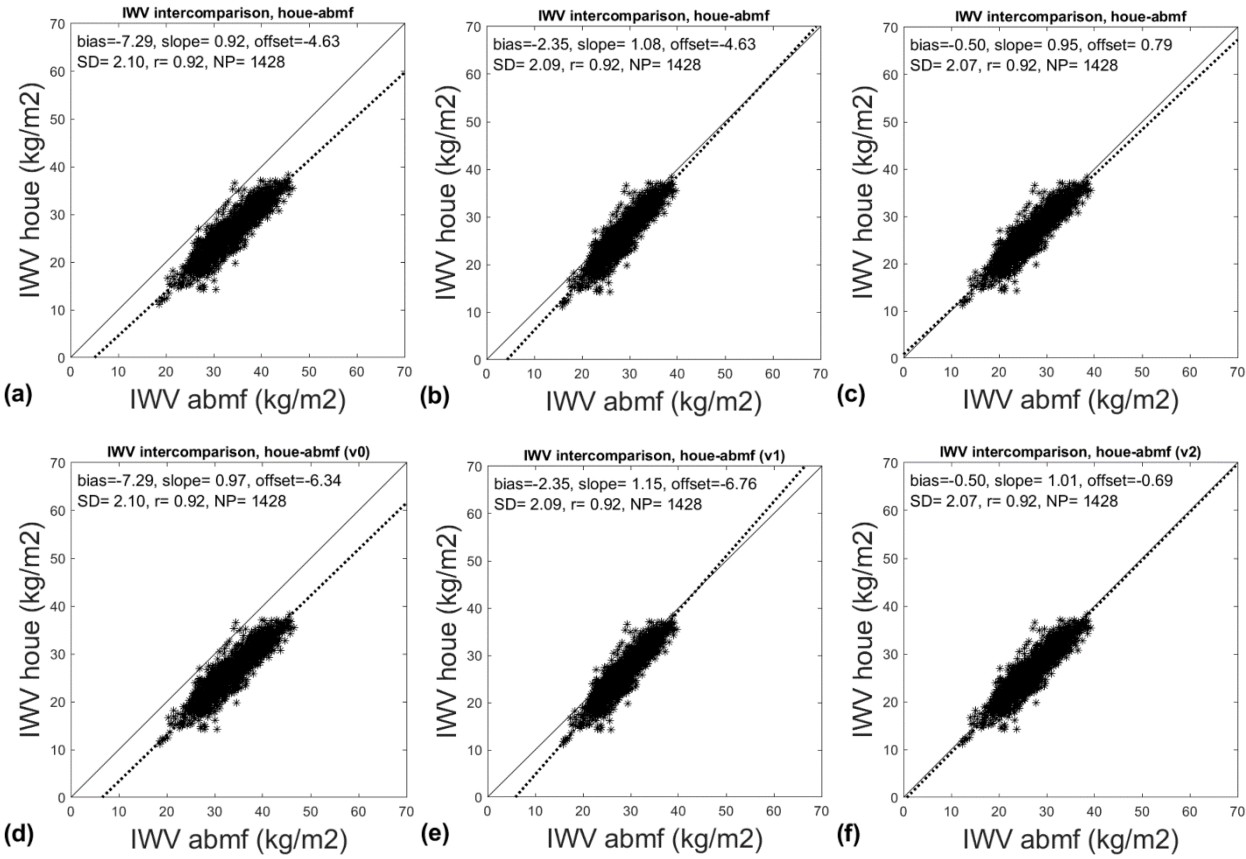

Figure 7: scatter plots of IWV observations from two GPS stations at different elevations (HOUE, 418 m, and ABMF, 15 m), before correction (a, d), after correction with a simple scaling factor model (b, e), and after correction with the proposed model fitted from a radiosonde climatology (c, f). The slope and offset parameters were either fitted by an ordinary least-squares method (a, b, c) or by the York et al. (2004) method accounting for errors in both coordinates (d, e, f). The data cover the period from 1 January to 29 February 2020, with a temporal resolution of 1 hour.

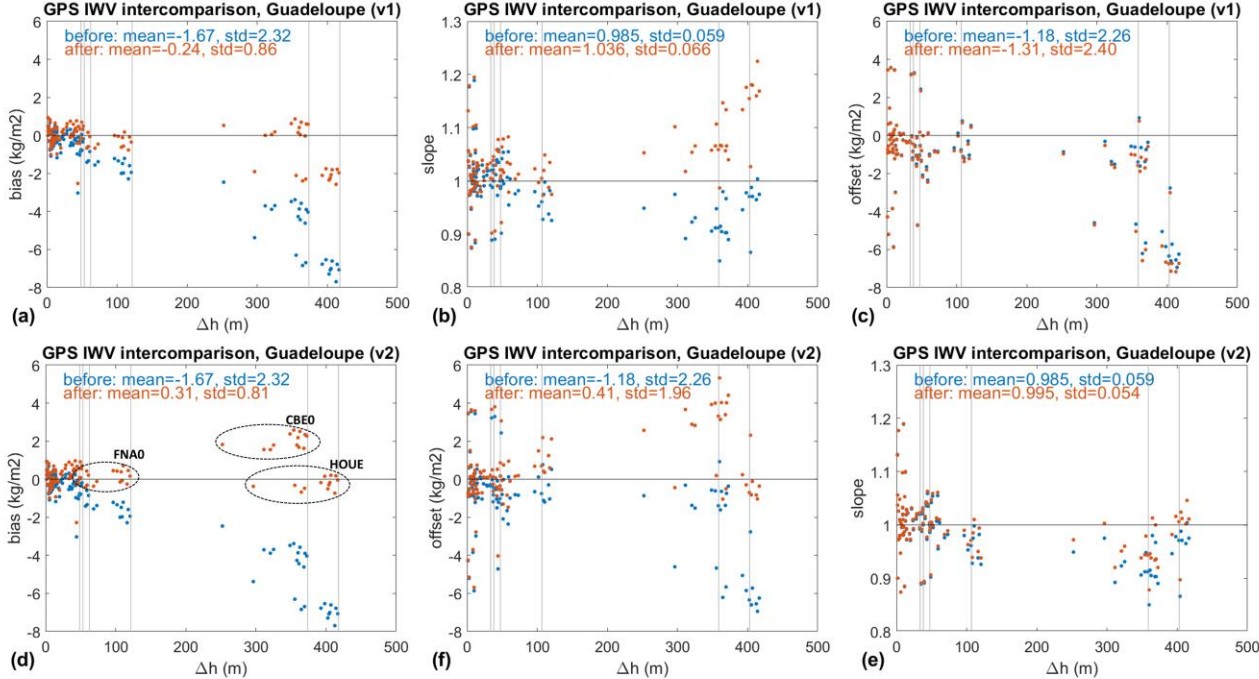


Figure 8: Variation of (a, d) bias, (b, e) slope, and (c, f) offset estimated from pairs of GPS stations as a function of between-station height difference, $\Delta h$. The blue dots correspond to the results before correction, and the red dots after correction, with: (a, b, c) the scaling model, v1, and (d, e, f) the proposed model fitted from a radiosonde climatology, v2. The results include 105 inter-comparisons with positive height differences, from a total of 15 GPS stations located over the Guadeloupe islands.

The grey vertical lines indicate the stations which have elevations above 50 m, namely: FNA0 (122 m), CBE0 (374 m), and HOUE (418 m), the comparisons of which are also highlighted by ellipses in Fig. 8d.

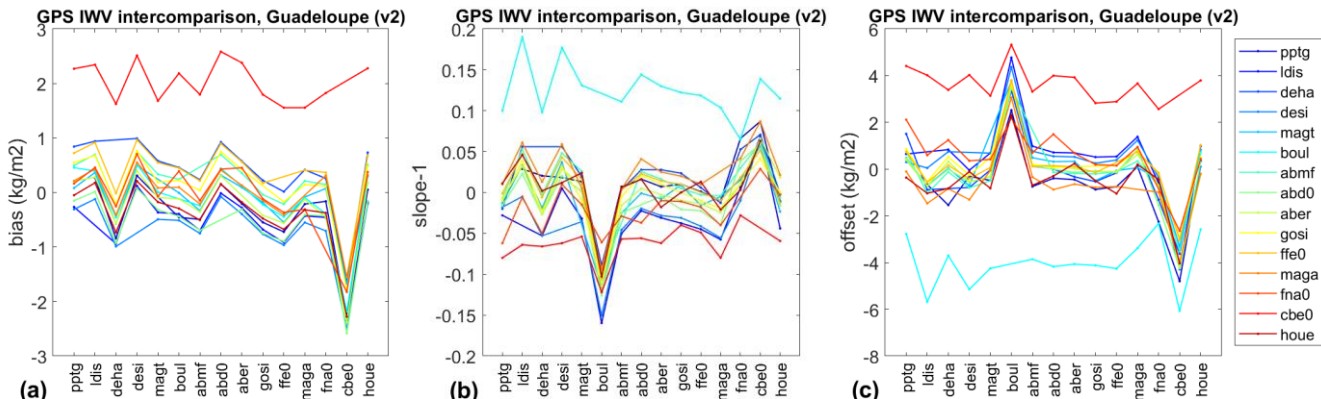


Figure 9: (a) bias, (b) slope – 1, and (c) offset estimated from all (210) pairs of GPS stations after correction with model v2. The station names along the *x*-axis refer to comparisons when the stations are in *x* while the colour code indicates comparisons when the stations are in *y*. The bias is always reported as $\Delta = \mu_y - \mu_x$ and the linear regression as $y = \alpha \cdot x + \beta$. For example, station CBE0 has a positive bias (red curve) when considered in *y*, while it has a negative bias when considered in *x*. The

comparison results from Fig. 8 were transformed using $\Delta' = \mu_x - \mu_y$, and $\alpha'$ and $\beta'$ according to Eqs. (B4a) and (B4b), when necessary.







Figure 10: Bias, slope, and offset results for GPS – GPS comparisons (blue line = median of all GPS comparisons from Fig. 9, except for the full year here) and for GPS – MWR comparisons from four different satellites (AMSR2, F18, GMI, and WINDSAT; black dashed line = mean of all satellite results), for all of year 2020. Slope and offset are estimated with York et al. (2004) method. Bias and offset values significantly different from 0, and slope values significantly different from 1, are marked with a circle (p-value ≤ 0.01). Pearson correlation coefficients between GPS – GPS and mean GPS – MWR results are indicated in the lower left angle of each plot.





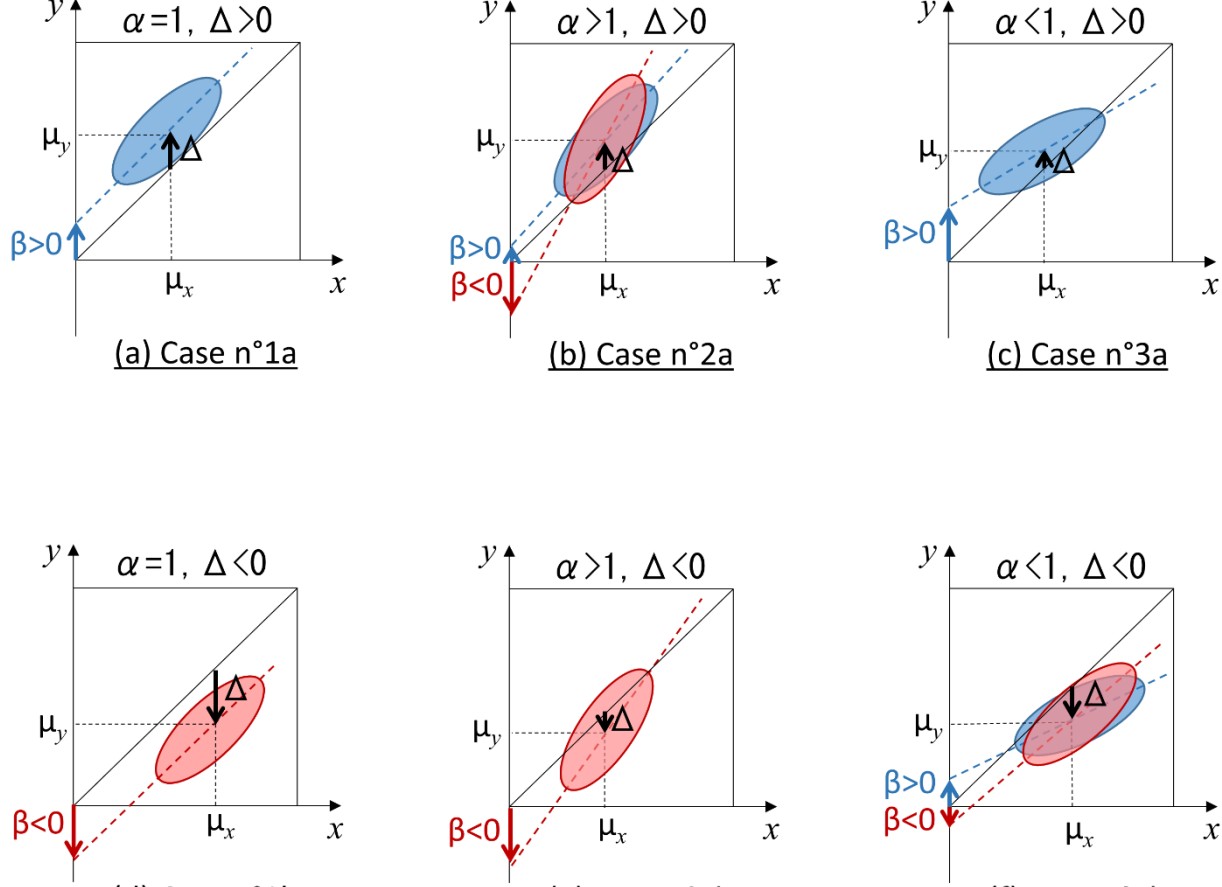

Figure B1: Illustration of different cases of paired observations with perfect scaling ($\alpha$ = 1), imperfect scaling ($\alpha$ > 1 or $\alpha$ < 1), and positive or negative bias ($\Delta > 0$ or $\Delta < 0$). Each case has a different implication on the offset parameter $\beta$ obtained from a linear regression with the model $y = \alpha x + \beta$. The regression lines are shown as dotted lines, with red colour indicating negative offsets and blue colour indicating positive offsets. The distributions of data around the regression lines are represented schematically by the red and blue ellipses.





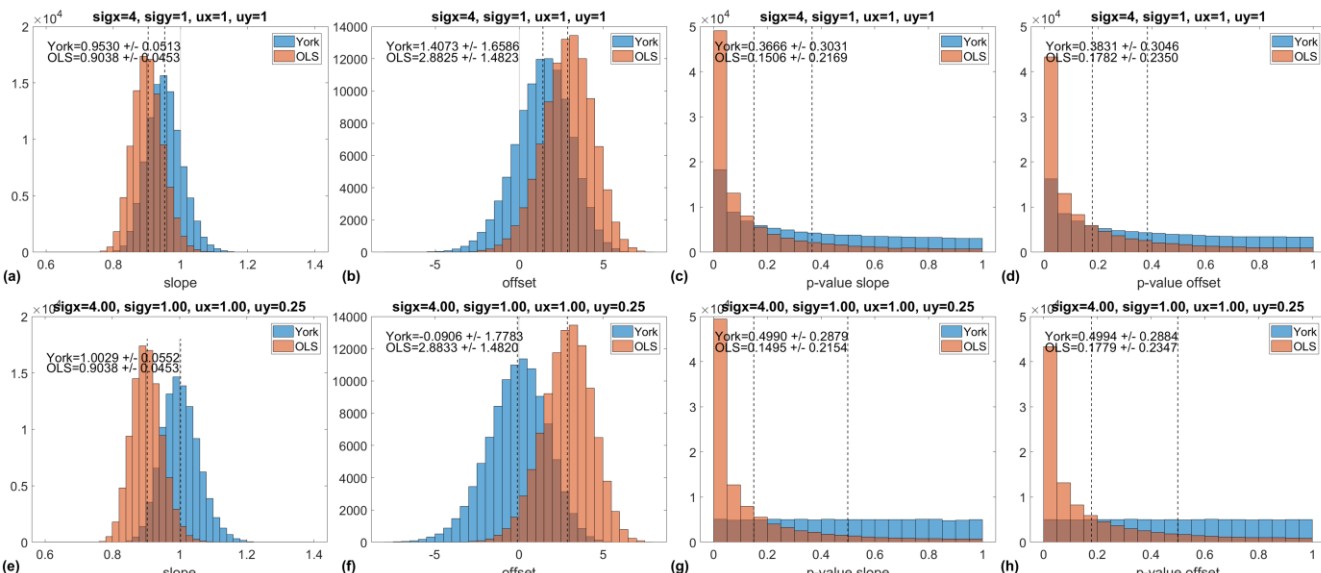

Figure C1: Monte Carlo tests of linear regression using the ordinary least-squares (OLS) and the York et al. (2004) method. The plots show the distributions of slope, offset, and respective p-values from the hypothesis tests ($H_0$: slope equal to one, and $H_0$: offset equal to zero, respectively). The dotted vertical lines indicate the mean values. Mean and standard deviation are reported in each plot. The true slope is 1 and the true offset is 0. (a-d) correspond to case n°5 from Table C1, and (e-h) correspond to case n°7. Each case is computed from $10^5$ simulations (see Appendix C for further details).