# Peer review of "An improved vertical correction method for the inter-comparison and inter-validation of Integrated Water Vapour measurements"

_Atmospheric Measurement Techniques, 2022_

## Author Comment (AC1)

Answers to referee comments on "An improved vertical correction method for the inter-comparison and inter-validation of Integrated Water Vapour measurements" by Olivier Bock et al., Atmos. Meas. Tech. Discuss., 2022.

Anonymous Referee #2

We thank the referee for the relevant comments and suggestions. The comments are repeated in black below, our answers are given in blue, and the corrections made in the manuscript are cited. A few supplementary figures are also included hereafter for further information.

1) One concern I have is that there might be problems related to the high polynomial order of the slope and offset parameters. The polynomial coefficients are derived using data in a specific height range (0 - 500m, 0-1000 m was also tested). Thus, there can be errors if the model is applied outside this range. This is especially true if the polynomial order is high. In the paper, all the GPS stations used are at heights below 500 m, so this is not a problem, but in general it can be. I think this issue should at least be discussed.

We totally agree with this comment. The model should not be applied outside the fitted range. This rule is well known in the regression context, but we added a note around Line 244 to recall it: "…vertical extent of the regression adapted to the application, i.e. verifying the condition that the fitted model is not used beyond the fitting range."

2) Are fifth degree polynomials really needed? Of course, the higher the degree, the better the fit to the data. However, there are some other reasons to keep the degree low. One is the above-mentioned problem when applying the model outside its fitted range. It is also easier to handle smaller number of coefficients, and the sensitivity to noise is lower. How much is actually gained in accuracy when using degree 5 instead of degree 2 or 3? Can it be quantified by some numbers?

As indicated on Line 212, we conducted a lot of trials for different values of the model parameters. The model actually fits better the data when higher order polynomials are used. Note that the regression method actually implements a stepwise procedure which selects the optimal order up to a pre-set maximum (but most of the cases the maximum is selected). In practice, the fine tuning of the model settings needs to take several factors into account such as the vertical range of the fit, the number of layers, the temporal sampling (we opted for monthly), and the desired accuracy of the final correction. We discussed these aspects a bit more in the revised manuscript (see Line 245 to 273). The choice of lower order polynomials is also quantified. For your information, Figure S1 below shows the results when we set the maximum order to 3. Compared to Fig. 6 of the paper, the results are slightly degraded but are still in an acceptable range.

3) At the end of Section 3, the parameters derived at three different radiosonde locations, separated by 400-500 km. are compared. It is found that they agree well and thus the parameters obtained from a radiosonde station can be applied to sites within several 100 km distance. I think this could be looked upon more closely. It is hard to draw any conclusions by just comparing the coefficients, it is difficult to know what is good a agreement and what is not. It might be better to look at what error in IWV would results if the model from one radiosonde site is applied at another one? Is it significant?

The coefficients compared in Fig. 4 of the paper show that the fitted coefficients are consistent, meaning that the atmosphere above the different sites has similar structure. However, we agree that it is difficult to translate this into IWV errors when the model from one station is used to correct the data from another station. Figure S2 below shows the results in such a case. Compared to Fig. 6 of the paper, the results are quite degraded (but note that the stations are 417 km apart). However, compared to the results of Fig. 8, it is clear that the new method would still achieve better results than the classical one (e.g., for a height difference around 400m, the bias is about -2 kg/m2, the slope close to 1.2 and the offset is not corrected with the classical method).

We also added a note in the manuscript on this point around Line 265:

"To quantify the impact of the second point, we used the model fitted for station 78954 to correct the data for station 78897. The resulting bias increased to 0.6 kg m-2."

4) Also, at the end of section 3 the difference between using monthly and yearly coefficients is investigated. It is found that it is better to use monthly coefficients, e.g., as seen in Fig 5. Can this be quantified by some numbers, i.e., the mean error when using monthly vs. yearly coefficients. Furthermore, if monthly is better than yearly, maybe it is even better going to even higher temporal resolution.

The difference between using monthly and yearly coefficients is quantified in Fig. S3 below. Compared to Fig. 6, the results are again degraded. We think that the monthly coefficients are a good compromise. They allow sampling properly the seasonal variation, which is the first order of temporal variation for IWV. Higher sampling rates, like e.g. daily or sub-daily, would probably not help much because day-to-day variations in IWV are actually dominated by changing weather, which is rather random in nature and not well predicted from a climatology.

5) The model coefficient for the traditional method, gamma=10^-4 m^-1, is taken from Bock et al 2007. I think it would be fairer to derive a new value from the radiosonde observations. Maybe even use monthly values, such as in the new method.

The value from Bock et al., 2007, is $4.10^{-4}$, actually, which is representative of the tropics. Deriving a new value is a subjective matter because one needs to choose a value that either better fits the bias or the slope. As we underline in the paper, with this model both cannot be fit properly simultaneously. This dilemma can be appreciated from Fig. 1 and 3. Figure 1b and c show that with the chosen coefficient, the bias at 1000 m is around -10 kg/m2 and the slope around 0.7. Figure 3a and b show that from the radiosonde data, the bias at 1000 m is between -15 and -18 kg/m2 and the slope between 0.85 and 0.93. Better fitting the bias would need to increase the coefficient value, but this would bring the slope further away from one. We think that from this perspective the chosen value is a good compromise.

6) For the GPS validation in Sec 4.1 it is not clear what time period was used. In the caption of fig. 7 it says 1 Jan- 29 Feb, 2020. Is this also true for the rest of the investigation in the section? In that case, why was only 2 months used instead of one full year (or a longer period)?

The GPS-GPS comparison was illustrated for 2 months only (Jan-Feb, 2020) because we were first interested in this period in the framework of the EUREC4A campaign (Bock et al., 2021). We added the information around Line 267 and in the captions of Figs. 8 and 9 in the corrected manuscript.

For the GPS-MWR comparison, we extended the analysis to the full year because the number of collocations over the 2-month period was a little too small to derive consistent statistics. We added a note on that around Line 365.

7) From the GPS validation, two problematic stations were identified, CBE0 and BOUL. In the validation by MWR, it is found that BOUL is no longer problematic. The reason is probably because of changes to this station made in 2020. It would be interesting to investigate this further to validate the assumption. One could look at different months and see for which months there are problems and for which there are not, and check if this agree whit the times changes were made to the station.

We inspected the monthly statistics for station BOUL from the GPS-GPS comparison (which have more data than the GPS-MWR comparisons). After correction with the new method, the slope parameter showed a drift throughout the year, from values > 1 in Jan-Feb (as seen in Fig. 9 of the paper) to values < 1 at the end of the year. This cannot be seen when all the data were regressed together, as in Fig. 10, because the average slope is close to 1. So it seems that it is not an abrupt change that is the cause of a

variation in the slope. Inspection of the station log information for that station also does not indicate an equipment change in 2020. We added a note on this around Line 370.

8) If in-situ meteorological measurements (humidity and temperature) are available, there are methods to calculate the height correction using these data. How does the new method compare to such methods?

It is not clear which methods and what kind of measurements (surface or upper-air) the reviewer is thinking of. If coincident upper-air measurements are available, the IWV contribution from the layer of interest could be readily calculated and this method would probably be more accurate than using a climatology. If only surface measurements are available, the method would need to make assumptions about the vertical decay. Some studies used, e.g., the power law approximation introduced by Smith, 1966. This method would probably have similar performance as the classical method discussed in our study, as both are based on a very similar concept, and so it would be inferior to the new proposed method.

Smith W. L. (1966) Note on the relationship between total precipitable water and surface dew point. J. Appl. Meteorol., 5, 726-727.

9) In the conclusion it is stated that the method can be applied to other regions where high resolution vertical water vapour profiles are available. What resolution would be needed. Would standard resolution radiosondes or typical resolution of numerical weather models be sufficient?

This is a good question. In the paper we used radiosonde profiles with a high vertical resolution which we limited to 25-m. As more and more operational radiosonde stations transmit their data in high resolution (BUFR format), a similar resolution could be easily implemented in other regions. Nevertheless, to answer the question, we tested the method with a vertical resolution degraded to 50 m and 100 m. This is also the typical vertical resolution that is available in numerical weather models. We found that in both cases the results are only very slightly degraded. Figure S4 shows the results for the case with 100 m. We added a note in the conclusion to mention this result around Line 453:

"A few additional trials showed that with a vertical resolution of 100 m, very good results are still achieved (e.g., bias error smaller than 0.1 kg m-2)."

[Figure]

Figure S1: similar to Fig. 6 of the paper but for p=q=3.

[Figure]

Figure S2: similar to Fig. 6 of the paper but with data from station 78897 corrected with model fitted from station 78954.

[Figure]

Figure S3: similar to Fig. 6 of the paper but when the yearly coefficients are used.

Figure S4: similar to Fig. 6 of the paper but when the model coefficients are fitted from radiosonde data with a vertical resolution degraded to 100 m and afterward interpolated for the correction of the high resolution data.

---

## Author Comment (AC2)

Answers to referee comments on "An improved vertical correction method for the inter-comparison and inter-validation of Integrated Water Vapour measurements" by Olivier Bock et al., Atmos. Meas. Tech. Discuss., 2022.

**Anonymous Referee #3**

We thank the referee for the comments and provide below a point by point answer. Referees' comments are repeated in black, and our answers are given in blue. The corrections made in the manuscript are also indicated.

1) You assume that the 'standard procedure' for the vertical interpolation follows the simple exponential law provided in the introduction of the manuscript, i.e., 2km scale height for IWV. Who defined this to be the 'standard procedure'? In literature (also see next comment) I find different 'standard procedures'. E.g. you may use weather model data, and calculate the interpolation coefficients or lapse rates from there. That's it. In fact in the end of the manuscript you mention that you are going to make use of ERA5.

Nowhere in the paper is this method referred to as the 'standard procedure'. We write that it is a "widely used one" (Line 48) and cite a few papers that used it. The 2-km scale height quoted in Appendix A is from the ITU 2017 reference standard atmosphere, but this value is just quoted as an example. Otherwise, empirical values determined from measurements by various authors are also given in the Introduction.

We agree that if coincident weather model data would be available, the correction could be directly calculated in this way. However, it is not clear a priori what vertical resolution from the model data is required to achieve a good correction. To provide a first insight, we tested the method with a degraded version of our radiosonde data. We found that for 50-m and 100-m resolutions, the results are only very slightly degraded (see also our answers to Referee #2). We added a note in the conclusion to mention this result around Line 453:

"A few additional trials showed that with a vertical resolution of 100 m, very good results are still achieved (e.g., bias error smaller than 0.1 kg m-2)."

Our idea with the ERA5 is rather to provide a global climatology of monthly correction coefficients rather than direct IWV corrections.

Note that one clear advantage of the proposed empirical correction method is to avoid the need for coincident, high resolution, weather model data.

2) Your procedure could be useful for the vertical correction of the so called zenith wet delay, right? In some processing schemes a priori zenith wet delays are applied (they are typically provided from gridded numerical weather model data) but a vertical correction is required. Can you comment on this in the introduction. Here are some useful papers:

Böhm, J., Möller, G., Schindelegger, M. et al. (2015) Development of an improved empirical model for slant delays in the troposphere (GPT2w). GPS Solut.

Dousa, J., and Elias, M. (2014), An improved model for calculating tropospheric wet delay, Geophys. Res. Lett.

Yes, the procedure would similarly apply to ZWD correction. We mentioned it in the Introduction of corrected manuscript around Line 41 and at the end of the Conclusions.